# LY6E mediates an evolutionarily conserved enhancement of virus infection by targeting a late entry step

Katrina B. Mar [1], Nicholas R. Rinkenberger[1], Ian N. Boys [1], Jennifer L. Eitson[1], Matthew B. McDougal [1], R. Blake Richardson[1] & John W. Schoggins [1]

Interferons (IFNs) contribute to cell-intrinsic antiviral immunity by inducing hundreds of interferon-stimulated genes (ISGs). In a screen to identify antiviral ISGs, we unexpectedly found that LY6E, a member of the LY6/uPAR family, enhanced viral infection. Here, we show that viral enhancement by ectopically expressed LY6E extends to several cellular backgrounds and affects multiple RNA viruses. LY6E does not impair IFN antiviral activity or signaling, but rather promotes viral entry. Using influenza A virus as a model, we narrow the enhancing effect of LY6E to uncoating after endosomal escape. Diverse mammalian orthologs of LY6E also enhance viral infectivity, indicating evolutionary conservation of function. By structure-function analyses, we identify a single amino acid in a predicted loop region that is essential for viral enhancement. Our study suggests that LY6E belongs to a class of IFN-inducible host factors that enhance viral infectivity without suppressing IFN antiviral activity.

[1] Department of Microbiology, University of Texas Southwestern Medical Center, 6000 Harry Hines Blvd, Dallas 75390 TX, USA. Correspondence and requests for materials should be addressed to J.W.S. (email: john.schoggins@utsouthwestern.edu)

Viral detection by the host cell triggers production of interferons (IFNs), a family of pro-inflammatory cytokines that contribute to the host antiviral response. IFN signaling activates the transcription of hundreds of IFN-stimulated genes (ISGs), some of which encode known effector proteins that inhibit various stages of the viral life cycle. Although the first antiviral ISGs were discovered several decades ago, effector characterization has so far been limited to a subset of proteins. Recent advances in systematic screening strategies have accelerated discovery of novel antiviral ISGs[1–4]. These screens have also revealed that a smaller subset of ISGs enhance viral infectivity through uncharacterized mechanisms.

We recently found that the ISG lymphocyte antigen 6 complex, locus E (LY6E, formerly RIG-E, SCA-2, TSA-1) enhances the infectivity of multiple enveloped RNA viruses[1,2,5]. LY6E belongs to the LY6/uPAR superfamily, which consists of multiple proteins containing eight to ten cysteines that form a highly conserved, three-finger folding motif through disulfide bonding[6]. The LY6/uPAR superfamily is diverse and includes numerous LY6 proteins, complement regulatory protein CD59, and lipoprotein binding protein GPIHBP1[7,8]. Like most LY6 family members, LY6E localizes to the cell surface via glycosylphosphatidylinositol (GPI) attachment[9]. Previous studies implicate LY6E in modulation of cell signaling[10–12], as well as a potential role in host susceptibility to viral infection[13–17]. LY6E has recently been shown to promote viral entry and replication of HIV-1[18] and an early step of the virus life cycle for West Nile virus, dengue virus, and Zika virus[19]. However, it remains unclear how LY6E enhances infectivity of other RNA viruses.

In the current study, we characterize the viral phenotype of LY6E. We show that LY6E enhances infectivity of multiple, enveloped RNA viruses in several cellular backgrounds. In mechanistic studies using influenza A virus as a model, we find that LY6E enhances viral uncoating after endosomal escape. Evolutionary analyses coupled with structure-function studies indicate conservation of enhancement by specific protein domains. We conclude that LY6E belongs to a growing class of IFN-inducible factors that broadly enhance viral infectivity in an IFN-independent manner.

## Results

**LY6E enhances a subset of enveloped RNA viruses.** In screens to identify ISGs that modulate viral infection, we previously showed that ectopic expression of human *LY6E* by lentiviral transduction enhanced the infectivity of multiple, genetically diverse viruses. These include members of the *Flaviviridae* (yellow fever virus (YFV), dengue virus (DENV), and West Nile virus (WNV)), *Togaviridae* (Chikungunya virus, O'nyong nyong virus (ONNV)), *Retroviridae* (human immunodeficiency virus (HIV-1)) and *Orthomyxoviridae* (influenza A virus (IAV), strain PR8) families[1,2,5]. Here, we confirmed that in immortalized human *STAT1*$^{-/-}$ fibroblasts, lentiviral-mediated LY6E expression enhanced the infectivity of the fluorescent reporter virus YFV-17D-Venus after 24 h, or approximately one replication cycle (Fig. 1a). We used *STAT1*$^{-/-}$ fibroblasts as these cells are unable to respond to IFN, thus allowing us to study LY6E independent of other IFN-inducible ISGs. To determine whether LY6E has effects beyond the first viral replication cycle, we infected LY6E-expressing cells with a low dose of YFV-17D-Venus and quantified the percentage of infected cells over time by flow cytometry. At every time point, we observed enhanced infection in LY6E cells as compared to control cells expressing either firefly luciferase (fluc) or empty vector control (Fig. 1b, c). The most striking difference in infection was a 300% increase observed at 36 h post-infection, after the onset of viral spread. At this time

point, we observed an increase in the percentage of infected cells expressing low levels of Venus, as reflected by mean fluorescence intensity (MFI) of the fluorescent reporter. Since newly infected cells express lower levels of viral antigens and virally encoded Venus, our data suggest that these cells were recently infected with virus that was at an early stage of replication. To more precisely determine the effect of ectopic LY6E expression on virus production, we quantified the amount of a non-reporter virus (YFV-17D) secreted into cell supernatants over time by plaque assay (Fig. 1d). We consistently found that compared to empty vector control cells, cells ectopically expressing LY6E produced at least twice as much virus. Together, these time course studies indicate that LY6E enhances infection within the first viral replication cycle, which correlates with more rapid viral spread and overall increased virus production.

We further examined the effect of LY6E expression on viruses representing diverse families. In corroboration with our screen, LY6E enhanced DENV and ONNV infection[1,5]. LY6E also increased infectivity of a non-reporter A/WSN/33 strain of IAV, the PRVABC59 strain of Zika virus, and a GFP reporter vesicular stomatitis virus (VSV, Fig. 1e). We further validated our previous screening data, suggesting that ectopic LY6E expression does not universally enhance infectivity of enveloped, single-stranded RNA viruses, as Sindbis virus (SINV), equine arteritis virus (EAV), and measles virus (MV) were not affected[2]. Additionally, we observed no enhancement of a replication-defective adenovirus serotype 5 vector (AdV5), which is a non-enveloped, double-stranded DNA virus (Fig. 1f). These results demonstrate that LY6E enhances infection of a subset of enveloped RNA viruses from diverse viral families.

**LY6E enhances infectivity of monocytic and fibroblast cells.** Because *LY6E* is ubiquitously expressed in multiple human tissues[20,21], we next sought to test whether ectopic LY6E expression enhanced viral infection in different cellular backgrounds. We observed a robust phenotype in wild-type (WT) immortalized human fibroblasts (HuFibr) and a moderate enhancement in a human osteosarcoma cell line (U2OS) (Supplementary Fig. 1a). The strong enhancement phenotype in both WT and *STAT1*$^{-/-}$ human fibroblasts indicates that the LY6E phenotype is independent of STAT1-mediated signaling. Ectopic LY6E expression in human lung adenocarcinoma cells (A549), human hepatoma cells (Huh7.5), human embryonic kidney cells (HEK293T), or Syrian baby hamster kidney cells (BHK) did not enhance YFV (Supplementary Fig. 1a). LY6E overexpression in the monocytic cell line THP-1 also enhanced susceptibility to IAV (Supplementary Fig. 1b). We examined basal and IFN-induced protein expression of LY6E to determine whether we could discern a pattern underlying cell type dependency. Basal LY6E expression is detectable by Western blot in *STAT1*$^{-/-}$ fibroblasts, HuFibr, U2OS, A549, and HEK293T. Basal expression is not observed in THP-1 or Huh7.5. Treatment with IFNα induced LY6E in THP-1, HuFibr, U2OS, and A549, but had no effect on protein expression in Huh7.5 or HEK293T. As expected, no induction was observed in *STAT1*$^{-/-}$ fibroblasts (Supplementary Fig. 1c). Overall, our data indicate that LY6E promotes viral infectivity in a cell type-specific manner, with the strongest phenotype in cells of fibroblast and monocytic lineages. This cell-type specificity appears unrelated to endogenous or IFN-inducible levels of LY6E.

**A related LY6/uPAR family member enhances YFV infection.** LY6E is the first member of the LY6/uPAR protein family that has been validated in several studies to enhance viral infection. We performed phylogenetic analysis of several LY6/uPAR family

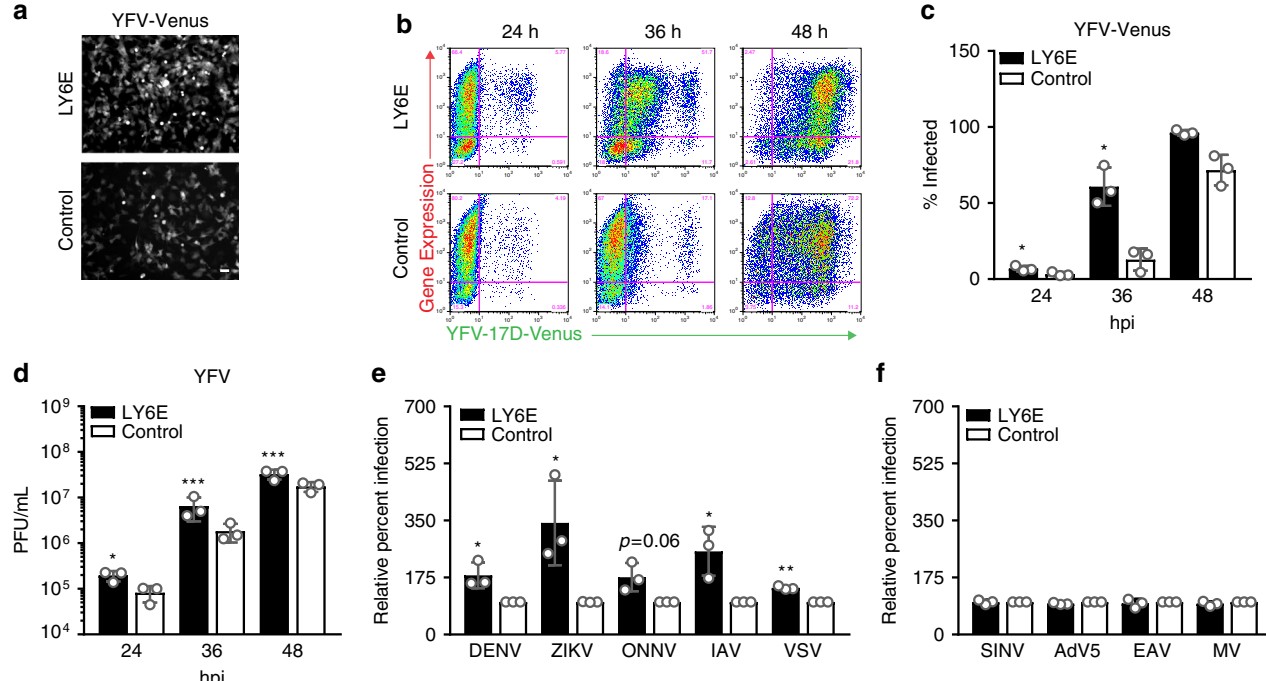

**Fig. 1** LY6E enhances infection by a subset of enveloped RNA viruses. **a** $STAT1^{-/-}$ fibroblasts transduced with lentivirus expressing LY6E or a fluc control were infected with 0.7 MOI YFV-17D-Venus and examined at 24 hpi by fluorescence microscopy. Scale bar, 100 μm. **b** $STAT1^{-/-}$ fibroblasts transduced with lentivirus expressing LY6E or a fluc control were infected with YFV-17D-Venus (0.04 MOI) and harvested for flow cytometry at 24, 36, and 48 hpi. Representative pseudocolor dot plots are shown. **c** Quantification of YFV infection as described in **b**. $n = 3$ biological replicates. **d** $STAT1^{-/-}$ fibroblasts transduced with lentivirus expressing LY6E or a fluc control were infected with YFV-17D (0.01 MOI). Supernatants were collected at 24, 36, and 48 hpi and titered on BHK cells by plaque assay. $n = 3$ biological replicates. **e** $STAT1^{-/-}$ fibroblasts transduced with lentivirus expressing LY6E or empty vector control were infected with DENV2-GFP (0.005 MOI, 48 h, control average 2.4% infection), ZIKV (PRVABC59, 1.2 MOI, 24 h, average 13.5% infection), ONNV-GFP (0.2 MOI, 17 h, control average 39% infection), IAV (A/WSN/33, 0.01 MOI, 8 h, control average 12.3% infection), or VSV (0.2 MOI, 5 h, control average 37.8% infection). Cells infected with IAV and ZIKV were permeabilized and respectively stained for NP or E protein. Percent infection was quantified by flow cytometry and is shown normalized to fluc or empty vector control. $n = 3$ biological replicates for each virus. **f** $STAT1^{-/-}$ fibroblasts transduced with lentivirus expressing LY6E or an empty vector control were infected with SINV-GFP (S300, 0.2 MOI, 10 h, control average 36.9% infection), AdV5-GFP (0.2 MOI, 24 h, control average 74% infection), EAV-GFP (0.8 MOI, 19 h, control average 22% infection), or MV-GFP (0.7 MOI, 24 h, control average 61% infection). Percent infection was quantified by flow cytometry and is shown normalized to fluc or empty vector control. $n = 3$ biological replicates for each virus. * $p < 0.05$, ** $p < 0.01$, *** $p < 0.001$. SD is shown. For **c**, **d**, **e** ratio paired $t$-test was performed to determine statistical significance. Raw data for **d** was log transformed before analysis. Statistical analysis of **e** was performed prior to normalization

member proteins of similar size and localization to LY6E (Fig. 2a, Table 1). While all LY6/uPAR family members share a highly conserved three-finger tertiary structure, only a maximum of 49.64% similarity in primary amino acid sequence is observed between LY6E and its nearest neighbor, prostate stem cell antigen (PSCA). To test whether other LY6/uPAR family members enhance viral infection, we generated lentivirus encoding LY6 family members and transduced $STAT1^{-/-}$ fibroblasts with a dose response of viral particles, with input normalized by p24 ELISA. Only PSCA enhanced infection of YFV-17D-Venus, while the expression of family members with less shared identity had no effect (Fig. 2b). Notably, PSCA was classified as a hit in a genome-wide siRNA knockdown screen to identify host factors important for WNV infection[13]. Our results suggest that a common sequence or structure shared by LY6E and PSCA, but not other family members, may underlie the viral enhancement phenotype.

**Loss of endogenous LY6E reduces viral susceptibility**. To investigate the role of endogenous LY6E in viral infectivity, we used CRISPR/Cas9 to genetically ablate protein expression. IFNα treatment induced expression of endogenous LY6E in THP-1 cells, but not in a bulk population of THP-1 transduced with Cas9 and a guide sequence targeting exon 2 of LY6E (Fig. 3a). Loss of LY6E protein expression in the absence of IFNα treatment greatly reduced susceptibility to IAV, indicating that basal protein levels

that are undetectable by Western blot can enhance viral infectivity (Fig. 3b).

We chose to further study endogenous protein in U2OS, which basally express high levels of LY6E without IFN treatment. We generated a clonal LY6E knockout (KO) cell line using CRISPR/Cas9 and confirmed by Sanger sequencing that each genomic LY6E allele contained nonsense mutations (Supplementary Fig. 2a). We observed complete loss of LY6E at the protein level in the KO cell line relative to WT U2OS (Fig. 3c). In the absence of LY6E, infectivity of IAV and YFV was reduced, but not completely abrogated (Fig. 3d). To complement the CRISPR-based LY6E deletion, we carried out a similar line of experiments using siRNA to silence LY6E expression (Supplementary Fig. 2b-e). Similar results were obtained, confirming that endogenous LY6E is not essential for viral infection, but is required for optimal viral infectivity.

**LY6E does not regulate IFN-mediated gene expression**. To assess whether LY6E could enhance viral infection via transcriptome modulation, we used RNA-Seq to compare gene expression profiles of $STAT1^{-/-}$ fibroblasts expressing an empty vector as a control versus cells expressing LY6E. Aside from ectopic LY6E expression, we observed no difference between the control and LY6E cells, indicating that LY6E does not influence the host transcriptome (Supplementary Fig. 3a).

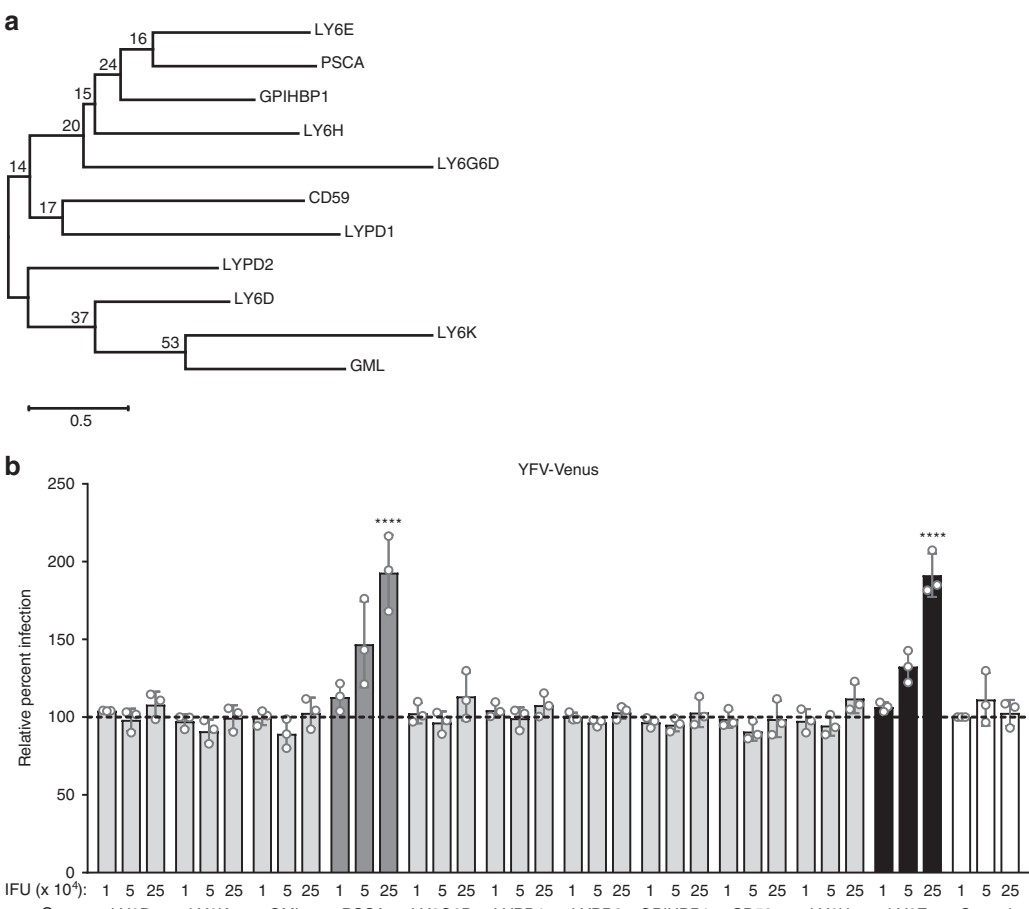

**Fig. 2** A related LY6/uPAR protein enhances yellow fever virus infection. **a** Molecular phylogenetic analysis by maximum likelihood method. The evolutionary history of LY6/uPAR family proteins was inferred by using the maximum likelihood method based on the JTT matrix-based model[64]. Bootstrap values resulting from 1000 replicates are shown next to the branches. The tree is drawn to scale, with branch lengths measured in the number of substitutions per site. Evolutionary analyses were conducted in MEGA7. **b** $STAT1^{-/-}$ fibroblasts transduced with three doses of lentivirus ($1 \times 10^4$, $5 \times 10^4$, $2.5 \times 10^5$ infectious units or IFU) expressing LY6/uPAR family members or fluc control were infected with YFV-17D-Venus (0.5 MOI, 24 h, control average 28.3% infection). Percent infection was quantified by flow cytometry and is shown normalized to cells transduced with $1 \times 10^4$ IFU fluc lentivirus. $n = 3$ biological replicates. **** $p < 0.0001$. SD is shown. Data were analyzed prior to normalization by ordinary two-way ANOVA with Dunnett's multiple comparisons test, which measured infectivity of LY6 family members relative to fluc control for each respective dose of lentivirus

| Table 1 LY6/uPAR family protein similarity | | | | | |
|---|---|---|---|---|---|
| Gene | Alignment length | Identical residues | Similar residues | Percent identity | Percent similarity |
| PSCA | 137 | 41 | 27 | 29.93 | 49.64 |
| LY6H | 144 | 39 | 24 | 27.08 | 43.75 |
| LY6D | 136 | 32 | 25 | 23.53 | 41.91 |
| GPIHBP1 | 192 | 45 | 21 | 23.44 | 34.38 |
| LYPD2 | 140 | 31 | 24 | 22.14 | 39.29 |
| LY6G6D | 140 | 27 | 23 | 19.29 | 35.71 |
| LYPD1 | 142 | 27 | 27 | 19.01 | 38.03 |
| CD59 | 139 | 26 | 21 | 18.71 | 33.81 |
| GML | 162 | 26 | 23 | 16.05 | 30.25 |
| LY6K | 169 | 26 | 24 | 15.38 | 29.59 |

Amino acid sequence identity and similarity of a subset of LY6/uPAR family members relative to human LY6E. The full sequence including the signal peptide and cleaved hydrophobic C-terminus was used for this analysis. The analysis was conducted using Sequence Manipulation Suite[67]

IFN induces the expression of negative regulatory genes, such as suppressor of cytokine signaling 1 (SOCS1), which can dampen antiviral IFN signaling and increase susceptibility to viral infection[22]. To determine whether LY6E modulates the type I

IFN response to enhance viral infectivity, we examined the effect of endogenous LY6E on the antiviral state induced by IFN treatment in LY6E KO and WT U2OS. Loss of endogenous LY6E did not alter the protective dose-dependent effect of IFN pre-treatment on YFV infection. Additionally, calculated $IC_{50}$ values were similar in both cell backgrounds (Supplementary Fig. 3b). At increasing doses of IFN, viral enhancement by endogenous LY6E was lost, likely due to IFN-mediated induction of antiviral ISGs that masked the enhancement phenotype. Accordingly, IFN-induced expression of antiviral ISGs previously shown to inhibit YFV was unchanged between LY6E KO and control cells, further demonstrating that LY6E-mediated viral enhancement is independent of the type I IFN response (Supplementary Fig. 3c)[1]. This data is corroborated by IFN-stimulated response element reporter studies[2]. Overall, our data suggest that LY6E acts in a direct manner to enhance viral infectivity, instead of in an indirect manner by influencing the cellular transcriptome or the IFN-mediated antiviral response.

**LY6E enhances yellow fever virus entry**. As a cell surface protein, LY6E may function as a viral attachment factor or receptor. To test this hypothesis, we incubated $STAT1^{-/-}$ fibroblasts expressing LY6E or empty vector control with increasing doses of

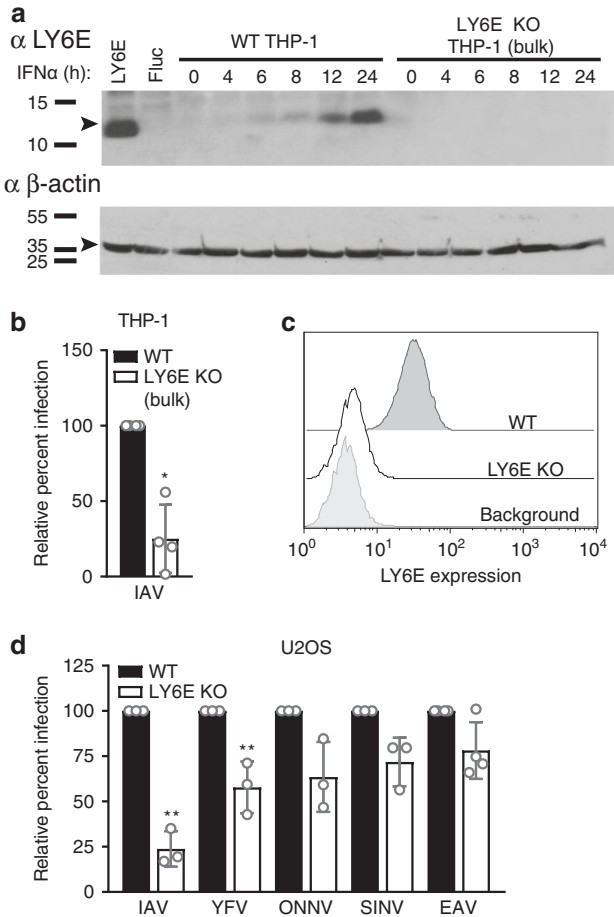

**Fig. 3** Loss of LY6E reduces but does not abrogate viral infection. **a** WT THP-1 cells and a bulk population of THP-1 expressing a LY6E-specific guide sequence and Cas9 (LY6E KO) were treated with 100 U/mL IFNα and harvested at indicated time points. LY6E and β-actin levels were assessed by Western blot. Lysates prepared from THP-1 ectopically expressing LY6E and fluc control constructs were used as positive and negative controls. Data are from one of three independent experiments. **b** Bulk LY6E KO and WT THP-1 were infected with IAV (A/WSN/33, 0.025 MOI, 8 h, WT average 37.7% infection), then permeabilized and stained for NP. Percent infection was quantified by flow cytometry and is shown normalized to infection of WT THP-1. n = 4 biological replicates. **c** Cell surface levels of endogenous LY6E in LY6E KO and WT U2OS. Cells were stained with anti-LY6E antibody and analyzed by flow cytometry. Background levels were determined by staining with an isotype control. **d** LY6E KO and WT U2OS were infected with IAV (A/WSN/33, 0.32 MOI, 8 h, WT average 27.9% infection), YFV-17D-Venus (0.6 MOI, 24 h, WT average 50% infection), ONNV-GFP (0.5 MOI, 17 h, control WT 18.9% infection), SINV-GFP (0.2 MOI, 10 h, control WT 14.6% infection), and EAV-GFP (0.8 MOI, 19 h, average 4.2% infection). Percent infection was quantified by flow cytometry and is shown normalized to infection of WT. n = 4 biological replicates for EAV and n = 3 biological replicates for IAV, YFV, ONNV, and SINV. * p < 0.05, ** p < 0.01. SD is shown. Data were analyzed prior to normalization by using student's unpaired t-test with Welch's correction

YFV-17D at 4 °C to block endocytosis and prevent viral entry. We observed no difference in bound viral RNA between LY6E and control conditions, indicating that LY6E does not affect YFV attachment to the cell surface (Fig. 4a).

To address the effect of LY6E on viral replication, we used a previously published subgenomic YFV-17D replicon assay[23]. In the replicon, the genes encoding structural proteins have been replaced with a *Renilla* luciferase (rluc) reporter gene. As a result, translation and replication of the subgenome can be inferred by quantifying rluc activity. The absence of structural proteins prevents the production of infectious virions, thereby decoupling viral entry and egress from genome translation and replication. STAT1−/− fibroblasts expressing LY6E or fluc control were transfected with replicon RNA, and luciferase activity was quantified at multiple early and late time points, which respectively correspond to viral translation and replication. LY6E expression did not promote YFV replicon translation (2–6 h) or replication (24–72 h), whereas IRF1 potently inhibited luciferase activity at all time points (Fig. 4b)[24].

To separate virus replication and production from earlier steps in the viral life cycle (attachment and entry), in vitro-transcribed viral RNA (YFV-17D) was electroporated directly into STAT1−/− fibroblasts expressing LY6E or empty vector control, bypassing endocytic entry pathways[25]. After 24 h, we measured replication by quantifying the MFI of YFV envelope protein (Fig. 4c). No difference was observed between STAT1−/− fibroblasts expressing LY6E or empty vector control, corroborating our observation with the subgenomic YFV replicon that LY6E does not enhance viral replication. Virus-containing supernatants were also harvested 24 h after electroporation and titer was determined by plaque assay. In contrast to the plaque assays presented in Fig. 1d in which supernatants were collected from cells that had been infected with intact YFV-17D virions, supernatants collected from STAT1−/− fibroblasts expressing LY6E or empty vector control that had been electroporated with YFV-17D RNA had equivalent viral titers (Fig. 4d). Therefore, in the absence of viral uptake, LY6E does not enhance production of YFV.

In Fig. 1a–c, we showed that LY6E enhances YFV-17D-Venus infection by increasing the percentage of infected cells. However, when viral entry is bypassed by direct electroporation of YFV-17D-Venus RNA into STAT1−/− fibroblasts expressing LY6E or empty vector control, we observe no difference in the percentage of Venus-positive cells (Fig. 4e). We conclude from these two experiments that the process of viral uptake is required for LY6E-mediated viral enhancement.

To further address the role of LY6E in viral entry, we used bafilomycin A1 (bafA), an inhibitor of endosomal acidification that blocks pH-dependent viral fusion[26]. YFV-17D-Venus virions that have already undergone endosomal fusion and escape before bafA treatment are able to replicate and express the Venus reporter. Because the number of internalized viral particles is limited by the capacity for attachment to the cell surface, we analyzed infection after 48 h to allow amplification of intracellular Venus expression for detection by flow cytometry. Addition of 5 nM bafA as early as 30 min after synchronized infection resulted in a greater percentage of Venus-positive cells in LY6E-expressing STAT1−/− fibroblast populations than in control fluc-expressing populations (Fig. 4f). However, Venus MFI was the same in STAT1−/− fibroblasts expressing LY6E as it was in cells expressing the fluc control, suggesting that viral uptake at the level of individual cells is similar in both cell backgrounds (Fig. 4g). Cumulatively, these mechanistic studies indicate that LY6E enhances YFV infectivity by affecting an early step in the viral life cycle that is after attachment to the cell surface but before viral translation, replication, and production.

**LY6E enhances influenza A virus uncoating**. To characterize the mechanism by which LY6E promotes infection of a virus unrelated to YFV, we next examined IAV. Using a similar cold-bind assay as for YFV, we observed that IAV attachment was not affected by LY6E expression (Fig. 5a). To evaluate the effect of LY6E on transcription and replication of IAV, we used a

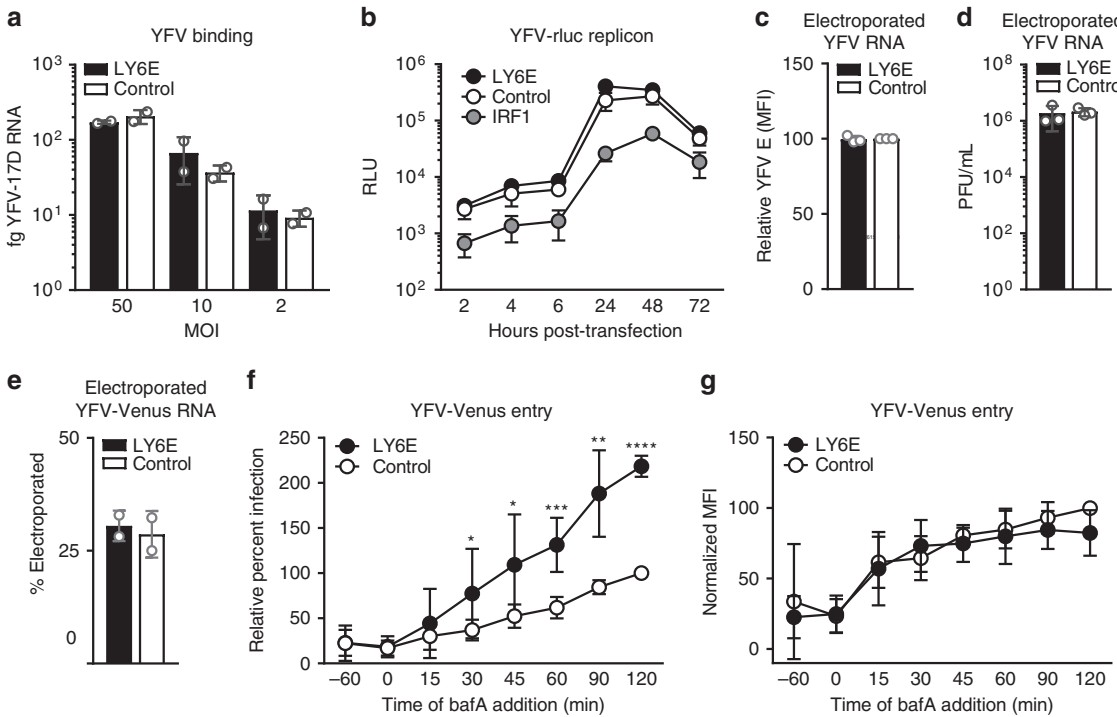

**Fig. 4** LY6E enhances entry of yellow fever virus. **a** STAT1[-/-] fibroblasts transduced with lentivirus expressing LY6E or an empty vector control were incubated with 50, 10, or 2 MOI YFV-17D at 4 °C for 1 h then harvested for qRT-PCR. $n = 2$ biological replicates. **b** STAT1[-/-] fibroblasts transduced with lentivirus expressing LY6E, IRF1, or a fluc control were transfected with YFV-17D-rluc replicon RNA. Lysates were assayed for *Renilla* luciferase activity at the indicated time points. Relative light units (RLU) are shown. $n = 3$ biological replicates. **c** STAT1[-/-] fibroblasts transduced with lentivirus expressing LY6E or an empty vector control were electroporated with YFV-17D RNA. Cells were permeabilized and stained for YFV E protein 24 h after electroporation. Replication was measured by flow cytometry as MFI of E protein. Infection is shown normalized to cells expressing empty vector control. $n = 3$ biological replicates. **d** STAT1[-/-] fibroblasts transduced with lentivirus expressing LY6E or an empty vector control were electroporated with YFV-17D RNA. Supernatants were collected 24 h after electroporation and titered on BHK cells by plaque assay. $n = 3$ biological replicates. **e** STAT1[-/-] fibroblasts transduced with lentivirus expressing LY6E or an empty vector control were electroporated with YFV-17D-Venus RNA. Percent electroporation was measured by flow cytometry as the percentage of cells that were Venus-positive. $n = 2$ biological replicates. **f** STAT1[-/-] fibroblasts transduced with lentivirus expressing LY6E or a fluc control were incubated at 4 °C for 1 h with YFV-17D-Venus (0.7 MOI). Virus was aspirated and cells were washed and shifted to 37 °C. BafA was added at indicated time points after the temperature shift. Cells were harvested 48 h after the temperature shift, and flow cytometry was used to quantify percent infectivity. Infection is shown normalized to fluc control at 120 min. $n = 4$ biological replicates. **g** Quantification of replication for **f**, as measured by MFI of the Venus reporter. $* \; p < 0.05, ** \; p < 0.01, *** \; p < 0.001, **** \; p < 0.0001$. SD is shown. In **b**, data for LY6E and control were log10 transformed then analyzed by unpaired *t*-test with Holm-Sidak correction. Ratio paired *t*-test was used to analyze **f**, **g** before normalization

previously published minigenome assay[27]. IAV polymerase activity was unchanged by the absence of LY6E in our clonal LY6E KO cell line relative to in WT U2OS (Fig. 5b). These data suggest that LY6E may enhance a post-attachment entry step of IAV, prior to the onset of replication.

IAV entry is composed of multiple steps: attachment, internalization, endosomal escape, capsid uncoating, and translocation of viral ribonucleoproteins (vRNP) into the nucleus. To determine if LY6E affects internalization of virus by endocytosis, we tagged IAV with a sulfo-NHS-SS-biotin tag, which allows detection of viral particles by using streptavidin conjugated fluorophores[28]. Treatment with the cell-impermeable reducing agent tris(2-carboxyethl)phosphine hydrochloride (TCEP) cleaves the disulfide bridge linker, efficiently removing the biotin tag from IAV attached to the cell surface but not from virus that has already been internalized. Infection of LY6E KO or WT U2OS with biotinylated IAV revealed equivalent levels of viral internalization (Fig. 5c).

After attachment and internalization, trafficking of endocytosed viruses through endosomal compartments results in exposure to low pH. For viruses that belong to the *Rhabdoviridae*, *Togaviridae*, *Flaviviridae*, and *Orthomyxoviridae* families, fusion of the viral envelope with the endosomal membrane is triggered by acidic pH[29]. To assess whether LY6E enhances endosomal escape, we used IAV labeled with lipophilic dye octadecyl rhodamine B (R18) at a concentration that results in self-quenching of the fluorescent signal. Upon fusion with the host endosomal membrane, R18 becomes diluted to the point of de-quenching, which results in a measurable fluorescent signal[28]. Pre-treatment with bafA, which blocks endosomal acidification, resulted in attenuated IAV fusion. Endosomal escape was equivalent in LY6E KO and WT U2OS, indicating that LY6E does not enhance endosomal escape of IAV (Fig. 5d).

Fusion of the viral envelope with host endosomal membranes leads to release of encapsidated viral genome into the cytosol. To assess whether LY6E affects a process downstream of endosomal escape, we used an acid bypass assay, whereby low pH triggers viral fusion at the plasma membrane, thus bypassing endosomal entry. Exposure of cells to acidic pH had no effect on LY6E-mediated enhancement of IAV, suggesting that LY6E affects steps downstream of endosomal escape (Fig. 5e). Performing the acid bypass assay in LY6E KO and WT U2OS corroborated the overexpression data, further indicating that LY6E expression enhances IAV entry after endosomal escape (Fig. 5f).

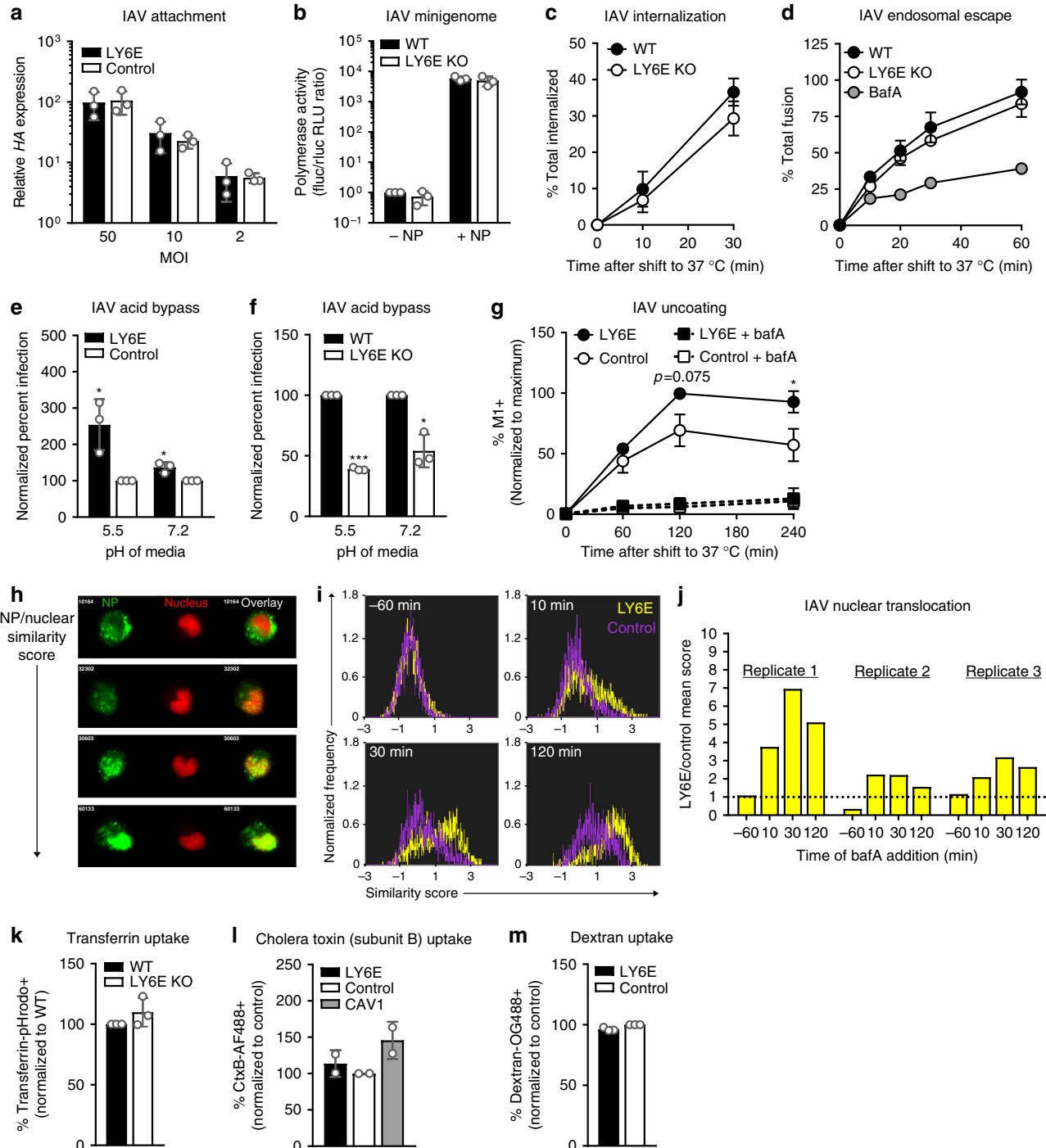

**Fig. 5** LY6E enhances entry of influenza A virus at the step of uncoating. **a** Cold-bind of IAV with *STAT1*[-/-] fibroblasts. HA expression is shown relative to that of control cells infected with 50 MOI IAV. $n = 3$ biological replicates. **b** IAV minigenome activity in U2OS cells. The fluc/rluc RLU ratio is shown normalized to the WT, no NP (-NP) condition. $n = 3$ biological replicates. **c** Internalization of biotinylated IAV with U2OS. Percentage of IAV internalization was calculated relative to the signal from cells not treated with TCEP. $n = 2$ biological replicates. **d** Endosomal escape of R18-labeled IAV with U2OS. Percentage of total fusion was calculated relative to maximal R18 dequenching. $n = 4$ biological replicates. **e** Acid bypass of IAV with *STAT1*[-/-] fibroblasts. Data are shown normalized to infection of control cells. $n = 3$ biological replicates. **f** Acid bypass of IAV with U2OS. Data are shown normalized to infection of WT cells. $n = 3$ biological replicates. **g** Viral uncoating as measured by M1 stain in *STAT1*[-/-] fibroblasts. Data are shown normalized to maximum percentage of positive cells within each experiment. $n = 3$ biological replicates. **h** Representative images of NP stain from *STAT1*[-/-] fibroblasts infected with IAV and treated with bafA at different time points by ImageStream. **i** Histograms indicating NP and nuclear signal overlay of IAV-infected cells as in **h** from replicate 1. **j** Fold difference of inverse log mean NP/nuclear similarity scores from *STAT1*[-/-] fibroblasts expressing LY6E relative to control. Scores are averaged from approximately $10^4$ cells per condition. **k** Transferrin uptake in U2OS. Data are shown normalized to WT control. $n = 3$ biological replicates. **l** Cholera toxin subunit B uptake in *STAT1*[-/-] fibroblasts. Data are shown normalized to fluc control. $n = 2$ biological replicates. **m** Dextran uptake in *STAT1*[-/-] fibroblasts. Data are shown normalized to fluc control. $n = 3$ biological replicates. * $p < 0.05$, ** $p < 0.01$, *** $p < 0.001$. SD is shown. Prior to data normalization, ratio paired *t*-test was used to determine statistical significance for **e**, **f**, **g**

Disassembly of the IAV capsid, or uncoating, can be monitored by staining for the M1 protein, which is a major component of the capsid. Dispersal of M1 in the cytosol increases accessibility to the monoclonal antibody HB64, resulting in a brighter cytoplasmic signal relative to the endosomal stain that is observed prior to uncoating[30,31]. To assess the effect of LY6E expression on uncoating, we incubated $STAT1^{-/-}$ fibroblasts with 25 MOI IAV at 4 °C to synchronize infection. After 1 h, unbound virus was aspirated and cells were shifted to 37 °C in the presence of cycloheximide to prevent translation of nascent M1 protein. As a control, we also treated cells with a combination of bafA and cycloheximide to block endosomal escape and thus prevent uncoating. Cells were harvested at 1, 2, and 4 h post-infection and stained with HB64. The percentage of cells with a bright M1 signal was assessed by flow cytometry. Ectopic expression of LY6E increased the percentage of M1-positive cells relative to control, indicating that LY6E promotes IAV uncoating (Fig. 5g). When both LY6E and control cells were treated with bafA, fewer cells were identified as M1-positive. This reduction in M1-positive cells is consistent with M1 antigen from acid-exposed, uncoated IAV being more accessible to HB64 than M1 antigen from endosomal IAV.

After endosomal escape and uncoating, the segmented genome of IAV translocates into the nucleus, where it is transcribed and replicated. Each genome segment is associated with nucleoprotein (NP), to form vRNP. By staining for NP and nuclei, we can distinguish cytoplasmic and nuclear vRNPs after addition of bafA (Fig. 5h). We batch-compared ten thousand infected $STAT1^{-/-}$ fibroblasts expressing LY6E or empty vector control for NP translocation into the nucleus. A low similarity score indicates cytoplasmic restriction of NP, whereas a high similarity score indicates NP has completely translocated into the nucleus. When cells were pre-treated with bafA, $STAT1^{-/-}$ fibroblasts expressing LY6E or fluc control had identical similarity scores, confirming that inhibition of endosomal acidification blocks viral escape and nuclear import of vRNP. When bafA was added as early as 10 min post-infection, a greater number of LY6E-expressing cells had nuclear vRNP as compared to control cells. By 2 h, most LY6E-expressing cells showed complete overlay of NP and nuclear signals, indicating that the LY6E enhancing effect results in increased NP import into the nucleus (Fig. 5i, j). Because vRNP translocation is downstream of viral uncoating, we hypothesize that the enhancement observed in the ImageStream assay is likely a result of the uncoating phenotype, though we cannot rule out that LY6E may also directly affect vRNP nuclear import. Further experimentation in which uncoating is uncoupled from vRNP nuclear translocation would be needed to distinguish these possibilities.

One common process shared by flaviviruses and IAV during viral entry is the use of clathrin-mediated endocytosis (CME)[32,33]. To test if LY6E affects uptake of non-viral particles, we used the classic CME ligand transferrin conjugated to the pH-dependent fluorophore pHrodo (Tfn-pHrodo). LY6E KO and control U2OS had a similar percentage of cells positive for Tfn-pHrodo, indicating that non-viral CME is not influenced by LY6E expression (Fig. 5k). Expression of LY6E also had no effect on caveolar endocytosis of cholera toxin, subunit B (Fig. 5l) or micropinocytosis of dextran (Fig. 5m).

Cumulatively, our data show that LY6E promotes viral entry by two enveloped RNA viruses from distinct families. While we were able to narrow the effects of LY6E to uncoating using IAV as a model, additional studies are needed to determine whether this mechanism applies to other viral families enhanced by LY6E expression.

**Viral enhancement by LY6E is conserved across evolution.** To gain insight into the significance of broad viral enhancement by

an IFN-inducible gene, we tested whether the LY6E phenotype was conserved across evolution. First, we performed phylogenetic analysis on LY6E orthologs from seven mammalian families (Supplementary Fig. 4a, Supplementary Table 1). While LY6E is well conserved in primates, we observed a 24% or greater decrease in amino acid identity and 15% or greater decrease in amino acid similarity when comparing human LY6E to non-primate orthologs. We tested for conservation of the LY6E-mediated viral enhancement phenotype by using orthologs representing diverse mammalian families. We expressed LY6E from Cercopithecidae (*Macaca mulatta* or rhesus macaque), Pteropodidae (*Pteropus alecto* or black flying fox), and Muridae (*Mus musculus* or house mouse) in human $STAT1^{-/-}$ fibroblasts and compared enhancement of YFV-17D-Venus to that of human LY6E. All orthologs enhanced infection by YFV-17D-Venus relative to the control when expressed in a heterologous cell type (Fig. 6a). Furthermore, expression of both human LY6E and the *M. musculus* ortholog Ly6e in murine fibroblasts (3T3) enhanced YFV-17D-Venus infectivity relative to the control (Fig. 6b). Expression of the murine ortholog in LY6E KO U2OS partially restored viral infectivity (Fig. 6c). Collectively, our analyses reveal that viral enhancement by LY6E is functionally conserved in diverse mammals.

**Leucine 36 is essential for viral enhancement by LY6E.** We hypothesized that the conservation of viral enhancement by LY6E orthologs may be due to sequence similarities. Initially we used an unbiased block mutagenesis approach in which stretches of approximately four amino acids within the mature LY6E protein were mutated to alanine (Fig. 7a, numbered blocks). The N-terminal signal peptide and post-GPI anchor hydrophobic C-terminus were not mutated for this experiment, as both segments are cleaved to produce the mature LY6E protein. Highly conserved cysteine residues contribute to disulfide bonding that is essential for the three-finger folding motif and were also not mutated. The block mutants were expressed in $STAT1^{-/-}$ fibroblasts by lentiviral transduction and tested for loss of LY6E-mediated viral enhancement. Alanine mutagenesis of blocks 1, 4, 5, 7, 8, 12, 13, 19, and 20 reduced YFV-17D-Venus infection to control levels, indicating either disruption of structure or of function (Fig. 7b). A partial reduction of viral enhancement was observed when blocks 10, 11, and 18 were mutated to alanine, suggesting that residues in these regions contribute, but are not essential, for the viral phenotype of LY6E.

For targeted structure-function analyses, we narrowed our mutagenesis to the predicted loop regions based off a structural model derived from the crystal structure of the three-finger protein irditoxin subunit B from Brown tree snake (Boiga irregularis, Supplementary Fig. 5a). Analogous regions of other LY6/uPAR family members have previously been shown to be functionally important for intermolecular interactions[34,35]. Because LY6E orthologs from mouse, rhesus macaque, and bat enhanced YFV-17D-Venus infectivity to the same effect as the human protein, we hypothesized that conserved residues shared by the four orthologs may be important for the viral phenotype. We identified nine loop residues of interest within blocks of amino acids that resulted in a loss of phenotype as shown in Fig. 7b, including blocks 4, 9, 10, and 11 (Fig. 7a and Supplementary Fig. 5a, arrows indicate mutated residues). The mutations were cloned into human LY6E with a HA tag added immediately before the GPI anchored serine at position 101 (S101), and the resulting HA-tagged mutants were expressed in $STAT1^{-/-}$ fibroblasts by lentiviral transduction. Input lentivirus was normalized between mutants by p24 ELISA. A partial loss of viral phenotype was observed with I57A. A complete loss of

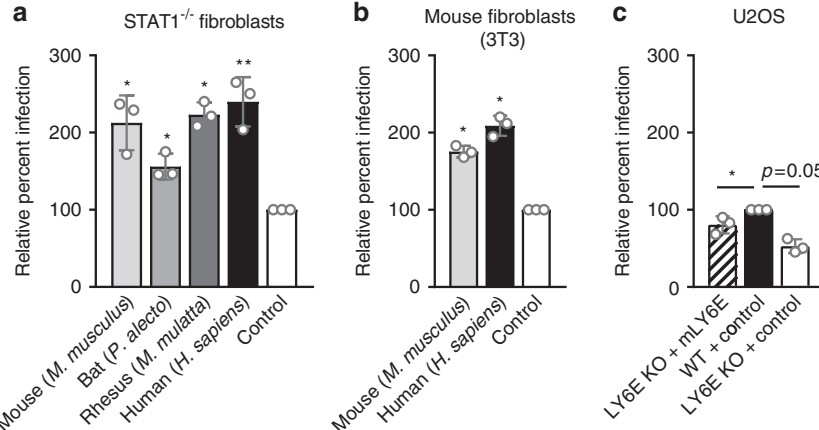

**Fig. 6** LY6E-mediated viral enhancement is conserved across mammalian species. **a** STAT1[-/-] fibroblasts transduced with lentivirus expressing human, rhesus, bat, or mouse LY6E orthologs or an empty vector control were infected with YFV-17D-Venus (0.25 MOI, 24 h, control average 13.1% infection). Percent infection was quantified by flow cytometry and is shown normalized to infection of control cells. $n = 3$ biological replicates. **b** Mouse fibroblasts (3T3) transduced with lentivirus expressing human LY6E, mouse Ly6e, or empty vector control were infected with YFV-17D-Venus (1 MOI, 24 h, control average 5.5% infection). Percent infection was quantified by flow cytometry and is shown normalized to infection of control cells. $n = 3$ biological replicates. **c** LY6E KO and WT U2OS transduced with lentivirus expressing mouse Ly6e or empty vector control were infected with YFV-17D-Venus (0.6 MOI, 24 h, control average 51.8% infection). Percent infection was quantified by flow cytometry and is shown normalized to infection of WT U2OS transduced with empty vector. $n = 3$ biological replicates. *$p < 0.05$, **$p < 0.01$. SD is shown. Statistical analyses were performed before data normalization using repeated measures one-way ANOVA, with the Greenhouse-Geisser correction and Dunnett's multiple comparisons test. Comparisons are relative to control for (**a**, **b**) and relative to WT + control for (**c**)

phenotype occurred with L36A, which fully recapitulated the loss of phenotype observed with mutagenesis of block 4 (Fig. 7c). To test whether these point mutations affected cell surface localization and protein expression, we used immunofluorescence and Western blotting to examine expression of the HA epitope. The point mutations had no effect on localization to the cell surface when compared to WT LY6E-HA (Supplementary Fig. 5b). Several mutations, including L33A, L36A, N59A, and G64A reduced protein expression as detected by Western blot for HA epitope (Supplementary Fig. 5c). Increasing protein expression of the L36A mutant by transducing with a greater amount of lentivirus did not rescue the viral phenotype relative to WT LY6E, indicating the loss of phenotype is not due to impaired LY6E expression or stability (Fig. 7d and Supplementary Fig. 5d). L36 was also required for enhancement of IAV, further demonstrating the importance of this residue (Fig. 7e). Overall, these structure-function analyses reveal L36 as an evolutionarily conserved residue that is important for LY6E expression and is essential for viral enhancement.

## Discussion

The ISG LY6E has previously been implicated by us and others to modulate viral infection. In the data presented here, we demonstrate that LY6E enhances viral infection in a cell type- and virus-specific manner. Using YFV, we show that LY6E expression promotes viral uptake, thereby increasing the percentage of cells in a heterogenous population that become infected (Fig. 4f). In the context of IAV infection, we show that LY6E promotes uncoating of IAV (Fig. 5g), though this mechanism of enhancement may differ for other viruses[18,19]. We further show that LY6E expression also increases the rate at which IAV enters cells as measured by NP translocation into the nucleus, which may be dependent on enhancement of uncoating (Fig. 5j).

What factors contribute to the specificity of LY6E-mediated viral enhancement? We demonstrate here that LY6E promotes infection by several viral families, but that the phenotype does not extend to all enveloped viruses. This selectivity may be due to differences in both entry strategies and dependence on host

machinery. As a small cell surface protein lacking a cytoplasmic tail, we hypothesize that LY6E requires interaction with other plasma membrane-associated proteins to confer activity. The flexible three-finger motif may allow multiple interacting partners depending on cellular context, as has been previously demonstrated with the T cell receptor in lymphocytes[36], nicotinic acetylcholine receptor[37], or syncytin-A[38]. Dependence on an interacting partner may also explain our observation in Supplementary Fig. 1a that the viral enhancement phenotype is absent in certain cell backgrounds. Furthermore, while we found LY6E expression to enhance VSV infection in STAT1[-/-] fibroblasts, another group observed inhibition of VSV when LY6E was expressed in HEK293T kidney cells[3]. We found that LY6E expression neither enhances nor inhibits YFV infection in HEK293T (Supplementary Fig. 1a), leading us to hypothesize that the viral phenotype of LY6E is variable depending on cell background and virus. This hypothesis may also help explain the diverse effects attributed to LY6E, such as attenuating T cell receptor signaling[10], suppressing responsiveness to LPS stimulation[14], or supporting proper placental formation[38,39]. As an IFN-inducible gene, the diverse functions of LY6E suggest that it may have pleiotropic effects that depend on cellular context.

LY6E may be one member of a growing class of ISGs that increases cellular susceptibility to viral infection, but whether the phenotypes in this class are all a result of viral hijacking remains unclear. Several ISGs that possess strong antiviral activity against diverse viruses can also be co-opted to promote infection by specific pathogens. IFITM3, which potently blocks entry of a broad range of viruses such as IAV, DENV, HIV-1, and Ebola virus (EBOV), can be used by human coronavirus OC43 as an entry factor[40,41]. Viperin, which can be induced by several viruses independently of IFN, blocks viral egress or replication of multiple viruses such as IAV, HIV-1, and DENV, but has also been shown to interact with human cytomegalovirus to enhance its infection[42,43]. ISGs with regulatory function but without antiviral activity have also been co-opted to promote infection of specific viruses, such as suppressor of cytokine signaling 3 (SOCS3), which can bind to EBOV protein VP40 to facilitate egress[44]. The

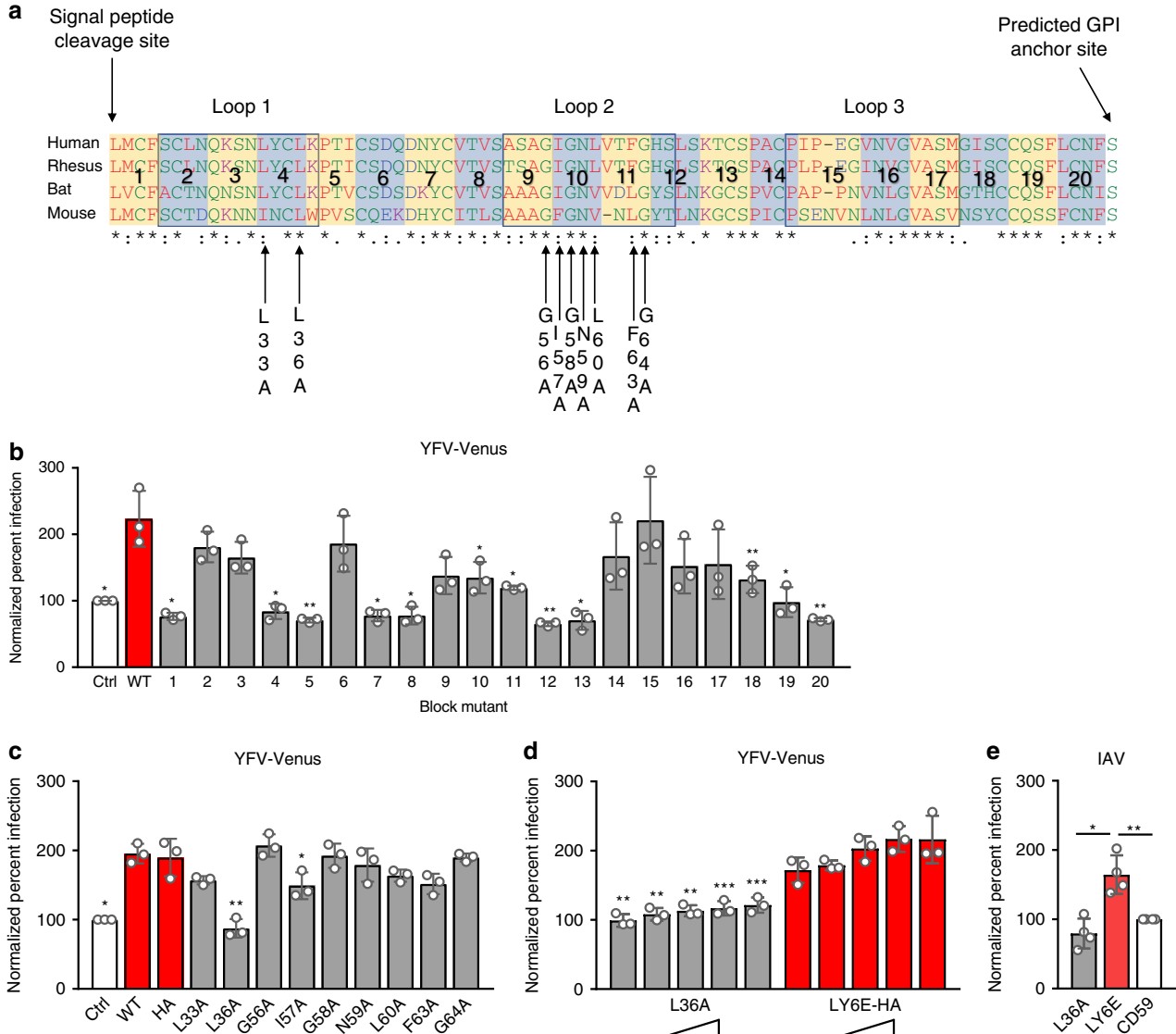

**Fig. 7** A conserved amino acid is essential for viral enhancement by human LY6E. **a** Alignment of amino acid sequences of mature LY6E protein orthologs. Red residues (AVFPMILW): small and hydrophobic or aromatic (except Y); blue residues (DE): acidic; magenta residues (RK): basic (except H); green residues (STYHCNGQ): have hydroxyl, sulfhydryl, or amine groups (plus G). Hyphens (-) indicate absence of a residue presents in other sequences. Consensus symbols indicate the following: *(asterisk) indicates a single, fully conserved residue, :(colon) indicates conservation between groups of strongly similar properties, .(period) indicates conservation between groups of weakly similar properties. **b** $STAT1^{-/-}$ fibroblasts expressing LY6E block mutants as shown in (**a**), WT LY6E, or fluc control were infected with YFV-17D-Venus (0.33 MOI, 24 h, control average 33% infection). Infection is shown normalized to fluc control. $n = 3$ biological replicates. **c** $STAT1^{-/-}$ fibroblasts expressing HA-tagged LY6E point mutant clones as shown in (**a**), LY6E-HA, or fluc control were infected with YFV-17D-Venus (0.25 MOI, 24 h, control average 16.4% infection). Infection is shown normalized to fluc control. $n = 3$ biological replicates. **d** $STAT1^{-/-}$ fibroblasts were transduced with five volumes of lentivirus (twofold increase per dose) expressing LY6E(L36A)-HA, LY6E-HA, or fluc control. Cells were infected with YFV-17D-Venus (0.25 MOI, 24 h, control average 24% infection). Infection is shown normalized to fluc control (not shown) for respective lentivirus doses. $n = 3$ biological replicates. **e** $STAT1^{-/-}$ fibroblasts expressing LY6E, LY6E (L36A), or CD59 were infected with IAV (A/WSN/33, 0.25 MOI, 8 h, control average 17.4% infection) and stained for NP. Infection is shown normalized to CD59 as a control. $n = 4$ biological replicates. * $p < 0.05$, ** $p < 0.01$, *** $p < 0.001$. SD is shown. **b**, **c** were analyzed before normalization by repeated measures one-way ANOVA with Greenhouse-Geisser correction followed by Dunnett's multiple comparison test relative to WT LY6E. For **d**, ratio paired $t$-test was used to compare L36A and LY6E at equivalent doses of lentivirus prior to data normalization. Data for **e** were analyzed prior to normalization by repeated measures one-way ANOVA with Holm Sidak's multiple comparison test relative to WT LY6E

receptor tyrosine kinase AXL is an example of an ISG that enhances infectivity by multiple viruses[45–48], but is also essential for cellular antiviral immunity by contributing to dendritic cell (DC) maturation. $Axl^{-/-}$ mice are more susceptible to IAV and WNV infection, indicating that the defect in DC maturation overshadows the effects of viral hijacking[49].

In an alternate model to viral hijacking, LY6E may serve to promote viral uptake in certain cell types to accelerate or amplify

innate and possibly downstream adaptive immune responses. This model would be consistent with data indicating that LY6E is ubiquitously expressed in multiple human tissues[21], and has been detected in multiple subsets of murine and human immune cells, both basally[9,50,51] and in response to IFN[14,20]. Identification of LY6E as a signature ISG that is upregulated in samples from IAV-infected patients suggests a potential role in host immunity[52]. Murine Ly6e was also identified by an in vivo RNAi screen as a

potential contributor to the murine antiviral response to IAV, making an intriguing case for further study of LY6E in the context of an intact immune system[53]. Our evolutionary analyses indicate functional conservation of the viral enhancement phenotype across distinct LY6E orthologs, which suggests a potentially beneficial role for the host. Whole body ablation of Ly6e has been reported to be embryonic lethal in mice due to defective placental formation[39,54]; therefore, we propose that mouse models in which Ly6e expression is ablated in specific cell types will be critical for testing complex models that are less tractable in cell culture systems.

## Methods

**Cell lines.** $STAT1^{-/-}$ fibroblasts, C8 human fibroblasts (HuFibr) and THP-1 were grown in RPMI (Invitrogen) with 10% FBS/0.1 mM non-essential amino acids (NEAA- Life Technologies). U2OS, A549, 3T3, Huh7.5, HEK293T and MDCK cells were grown in DMEM (Invitrogen) with 10% FBS/0.1 mM NEAA. BHK-21J and Vero cells were respectively maintained in MEM (Invitrogen) supplemented with 10% FBS/0.1 mM NEAA and OptiPRO serum-free medium supplemented with 4 mM glutamine (Gibco). $STAT1^{-/-}$ fibroblasts, C8 fibroblasts, BHK-21J, 3T3, U2OS, THP-1, Huh7.5, HEK293T and A549 stable cell lines were maintained with the addition of 4 µg/mL puromycin. U2OS and THP-1 KO cell lines were generated through transduction with lentiCRISPRv2-containing single guide RNAs predicted to target exons of LY6E and selected for 21 days in 2–4 µg/mL puromycin[55]. For LY6E KO in U2OS, a single-cell clonal population was isolated by FACS (MoFlo, UTSW Flow Cytometry Facility Core) and subsequently validated and expanded. The clonal population was confirmed by isolation of genomic DNA and Sanger sequencing. $STAT1^{-/-}$ and C8 fibroblasts were a kind gift from J.-L. Casanova. MDCK, BHK-21J, Huh7.5, Vero, and HEK293T cells were a kind gift from C. Rice. 3T3 mouse fibroblasts cells were a kind gift from M. Diamond. Other cell lines were obtained from the ATCC [THP-1 (cat# TIB-202), U2OS (cat# HTB-96), A549 (cat# CCL-185)].

**Viruses.** YFV-17D-Venus and YFV-17D were generated by electroporating in vitro transcribed viral RNA into BHK-21J or $STAT1^{-/-}$ fibroblasts cells as described previously[56]. DENV2-GFP was generated by electroporating in vitro transcribed T7-RNA into Huh7.5 cells as described previously[5]. IAV (A/WSN/33) virus was generated by inoculation of sub-confluent MDCK per protocol[57]. MV-GFP, SINV-GFP, EAV-GFP, and ONNV-GFP were generated as previously described[2]. ZIKV-PRVABC59 was generated as previously described[58]. AdV5-GFP was kindly provided by Robert Gerard and propagated in E1A-complementing 293 cells. VSV-GFP was produced in BHK cells. Concentrated IAV was obtained per protocol with several modifications[59]. In brief, viral supernatant from infected MDCK was centrifuged once at $2000 \times g$ for 30 min at 4 °C and then subjected to ultracentrifugation (Sorvall Discovery 100SE, SW28 rotor) for 30 min at $18,000 \times g$ at 4 °C to pellet cell debris. Virions from the resulting supernatants were pelleted through a 5 mL 30% sucrose cushion by ultracentrifugation for 90 min at $112,000 \times g$ at 4 °C. Viral pellets were resuspended in a small volume of 1X NTC (0.1 M NaCl/0.02 M Tris-HCl, pH 7.4/5 mM $CaCl_2$) and then diluted into a larger volume of 1X NTC. Virus was finally pelleted by ultracentrifugation at $154,000 \times g$ (SW40 rotor) for 60 min at 4 °C. Supernatant was aspirated and IAV was resuspended in a small amount of 1X NTC and titered in MDCK by plaque assay.

**In vitro transcription of viral and replicon RNA.** The following viral/replicon RNA were in vitro transcribed with mMessage mMachine SP6 kit (Ambion): YFV-17D-Venus RNA from XhoI-linearized YF17D(5′C25Venus2AUbi) plasmid[60], YFV-17D from XhoI-linearized pACNR-17D-Yfx[61], YFV-R.luc2A-RP replicon RNA from XhoI-linearized YFV-R.luc2A-RP[23]. The following RNAs were in vitro transcribed with mMessage mMachine T7 kit (Ambion): DENV2-GFP RNA from XbaI-linearized pDV2.IC30P.A.eGFP.P2AUbFIX plasmid[5]. RNA was purified from the transcription reaction using RNeasy mini kit (Qiagen) and quantified by Nanodrop.

**Pseudoparticle generation.** Lentiviral pseudoparticles using the TRIP lentiviral backbone were generated as previously described[1]. Stable cell lines were generated using the SCRPSY lentiviral backbone as described previously[5]. When stated, p24 ELISA (Clontech) was used per manufacturer's instructions to normalize lentiviral input.

**Transductions, infections, and plaque assays.** One day before transduction $7 \times 10^4$ cells were plated into 24-well plates. The next day, cells were transduced with lentiviral pseudoparticles by spinoculation at $800 \times g$, for 45 min at 37 °C in DMEM or RPMI containing 3% FBS/0.1 mM NEAA/20 mM HEPES/4 µg/mL polybrene. Six hours after transduction, media was changed to 10% FBS/0.1 mM NEAA/RPMI or DMEM. THP-1 cells were transduced with RPMI containing 3% FBS/0.1 mM NEAA/20 mM HEPES/10 µg/mL polybrene and media was changed immediately

after transduction to 10% FBS/0.1 mM NEAA/RPMI. Forty-eight hours post transduction SCRPSY stable cell lines were pooled in a 10 cm dish and allowed to reach confluency prior to addition of puromycin (titrated for each cell line, between 2–8 µg/mL). For transient transductions with TRIP construct lentiviruses, cells were split 1:3 ($7–10 \times 10^4$ cells) for infection approximately 48 h post-transduction and infected the following day.

Infections were carried out in a minimum volume of RPMI or DMEM with 1% FBS for 1 h (all viruses except IAV, MV, ONNV), followed by aspiration of virus and addition of 10% FBS/0.1 mM NEAA/RPMI or DMEM. Infection with IAV was carried out in a minimum volume of 0.3% BSA/0.1% FBS/PBS++ for 1 h, followed by aspiration of virus and addition of 0.3% BSA/0.1% FBS/0.1 mM NEAA/1 µg/mL TPCK/RPMI or DMEM. Infections with MV were carried out for 2 h in 1% FBS/ media, followed by addition of complete media. Infections with ONNV were carried out for 1 h in 1% FBS/media followed by addition of complete media. Adherent cells (all but THP-1) were dissociated with 200 µL Accumax (Sigma, diluted 1:4 in PBS for some $STAT1^{-/-}$ fibroblast infections) and transferred to a 96-well V-bottom plate. Cells were pelleted at $800 \times g$ for 2 min at 4 °C and resuspended in 1% PFA for 10 min at room temperature or 30 min at 4 °C. Cells were pelleted and resuspended in 3% FBS/PBS and kept at 4 °C until FACS analysis. IAV-infected and ZIKV-infected cells were permeabilized and respectively stained with a primary antibody reactive to IAV NP (1:500-1:1000 anti-NP, MAB8251, Millipore) or with a primary antibody reactive to flavivirus E protein (1:1000 anti-E, D1-4G2-4-15, Millipore) and a goat anti-mouse secondary IgG antibody conjugated to AlexaFluor488 using Cytofix/Cytoperm Kit (BD Bioscience) per protocol. Samples were run in a Stratedigm S1000 flow cytometer with a A600 96-well plate high throughput extension and compensated using CellCapture software (Stratedigm). Data was analyzed with FlowJo software (Treestar). The flow gating strategy for infection with GFP-expressing viruses of cell lines expressing RFP and a gene of interest is included in Supplementary Fig. 8a.

For YFV plaque assays, $STAT1^{-/-}$ fibroblasts were infected with YFV-17D for 1 h. Viral supernatant was then removed and cells were washed twice with serum-free RPMI. A total of 500 µL complete medium was added to wells and harvested 24, 36, and 48 h post-infection and stored at −80 °C. Collected supernatants were thawed and diluted, then used to infect BHK-21J cells. After 1 h infection with viral supernatants, 3 mL of overlay medium (0.1% $NaHCO_3$/4% FBS/10 mM HEPES/1.3% Avicel/1000 U/mL penicillin/1000 µg/mL streptomycin/1X DMEM) was added to infected cells and allowed to incubate for 4 days at 37 °C. After 4 days, 37% PFA was added to overlay medium/viral supernatant to fix cells. The supernatant was then aspirated and wells were stained with crystal violet to visualize plaques.

A similar method was used for titering concentrated IAV preps. In brief, confluent layers of MDCK were infected with concentrated IAV diluted tenfold for 1 h at 37 °C. Viral supernatant was aspirated and 3 mL of flu overlay medium (0.1% $NaHCO_3$/0.3% BSA/1% Avicel/1000 U/mL penicillin/1000 µg/mL streptomycin/2 µg/mL TPCK/1X DMEM) was added to infected cells and allowed to incubate for 2 days at 37 °C. Cells were fixed and stained as described above to visualize plaques.

**LY6E antibody staining.** Cells were harvested with Accumax and fixed in 1% PFA as described above. Cells were then washed once in 3% FBS/PBS before incubating with primary antibody (1:250 anti-LY6E, HPA027186, Sigma, discontinued) for 30 min at 4 °C. Cells were washed twice with 3% FBS/PBS before incubation with goat anti-rabbit secondary IgG conjugated to AlexaFluor594 for 30 min at 4 °C. Incubation was followed with two washes and resuspension in 3% FBS/PBS pending FACS analysis.

**Western blotting.** Samples were washed once in 1X PBS, resuspended in cold RIPA (50 mM Tris, pH 8.0/150 mM NaCl, 0.1% SDS/2 mM EDTA/0.5% sodium deoxycholate/1% NP-40/1X cOmplete Protease Inhibitor Cocktail Tablets [Roche]), and sonicated at 20% power for $2 \times 10$ s (Sonics Vibra-Cell Model CV188) Samples were subject to centrifugation at $10,000 \times g$ for 10 min to remove cellular debris. The protein concentration of the remaining supernatants was quantified by bicinchoninic acid assay (Pierce), normalized to a BSA standard curve, on a LUMIstar OPTIMA Microplate Reader (BMG LABTECH). 1X SDS loading buffer (63 mM Tris-HCl, pH 6.8/0.25% SDS/0.0025% bromophenol blue/10% glycerol/5% β-ME) was added and samples were boiled at 95 °C for 5 min and frozen at −80 °C. THP-1 lysates (25 µg) were thawed and loaded into a low molecular weight tricine gel (running gel: 10% acrylamide/13.3% glycerol/0.1% SDS/1.0 M Tris base, pH 8.45/0.05% ammonium persulfate[AMPS]/0.05% tetra-methylethylenediamine[TEMED]; stacking gel: 4% acrylamide/0.07% SDS/0.7 M Tris base, pH 8.45/0.08% AMPS/0.08% TEMED) and ran at 75–150 V in running buffer (0.1 M Tris/0.1 M tricine/0.1% SDS, pH 8.25 [inner chamber] and 0.2 M Tris, pH 8.9 [outer chamber]). Protein was transferred to PVDF membrane at 100 V for 40 min at 4 °C in transfer buffer (25 mM Tris/192 mM glycine/20% methanol). Membranes were blocked in 5% milk/TBST (10 mM Tris, pH 7.5/50 mM NaCl/0.1% Tween-20) for 30 min at room temperature before overnight incubation at 4 °C in 1:1000 αLY6E (a kind gift from Jyoti Asundi, Genentech)/1:40,000 αβ-actin (ab6276, Abcam) diluted in 5% milk/TBST. Membranes were washed $3 \times 5$ min in TBST then incubated for 30 min at room temperature with 1:2000 goat anti-mouse IgG conjugated to HRP and diluted in 5% milk/TBST. Membranes were

again washed 3 × 5 min in TBST before ECL and preparation for film exposure and development. Uncropped blots are included in Supplementary Fig. 6a.

**YFV cold-bind assay.** One day after plating, transduced cells were equilibrated to 4 °C for 30 min in complete medium. Media was then aspirated and YFV-17D diluted in cold 1% FBS/RPMI was added and incubated for 1 h at 4 °C. Cells were then washed 2 x with ice cold 1X PBS and harvested for RNA by RNeasy Mini Kit (Qiagen). Viral concentration was quantified by qRT-PCR with published primers[62]. A standard curve was generated by spiking in vitro transcribed YFV-17D-Venus RNA into a background of 40 ng uninfected cellular RNA and used to back-calculate fg YFV RNA for each sample based on $C_T$ value.

**YFV replicon assay.** Transduced cells were plated at 3.5 x $10^4$ cells per well in a 48-well plate the day before transfection. Viral replicon RNA was transfected into cells using the TransIT mRNA Transfection Kit (Mirus Bio) with a modified protocol (0.065 µg viral RNA, 0.25 µL mRNA Boost Reagent, 0.375 µL TransIT-mRNA reagent, and 25 µL serum-free Optimem [Gibco]). Renilla luciferase activity was quantified per protocol using the Renilla Luciferase Assay System (Promega) and LUMIstar OPTIMA Microplate Reader (BMG LABTECH).

**YFV electroporation assay.** STAT1[−/−] fibroblast cell lines expressing LY6E or vector control were pelleted, washed twice with ice cold 1X PBS, counted, and diluted to 1.5 × $10^7$ cells/mL in ice cold 1X PBS. Cells (6 × $10^6$) were electroporated (BTX-Harvard Apparatus ECM 830 Square Wave Electroporator) with 5 pulses of 860 V, at 99 us, 1.1 s intervals with 7.5 µg YFV-17D or YFV-17D-Venus RNA. After 10 min, cells were plated on a 100 cm² tissue culture plate. Media was changed 6 h post-electroporation to complete media. Supernatant and cells were harvested 24 h later. Cells were washed, resuspended in PBS, stained for YFV E protein (4G2, Millipore MAB10216) and harvested for FACS analysis as described above. Supernatant was used to infect BHK-21J for plaque assay as described above.

**YFV bafilomycin A1 entry time course.** Transduced STAT1[−/−] fibroblasts were plated at 7 x $10^4$ cells per well in a 24-well plate the day before infection. Plates were equilibrated to 4 °C on ice prior for 30 min to addition of YFV-17D-Venus diluted in cold 1% FBS/RPMI. Cells were cold-bound for 1 h at 4 °C and then washed twice with cold 1X PBS. Warm complete RPMI was then added and cells were shifted to 37 °C. Bafilomycin A1 (Sigma) was added to cells at indicated time points for a final concentration of 5 nM. Infections proceeded for 48 h before cells were harvested for flow cytometry.

**IAV cold-bind assay.** One day after plating, transduced cells were equilibrated to 4 °C for 30 min in complete medium. Cells were incubated for 1 h at 4 °C with IAV (A/WSN/33) at the indicated MOI in cold 0.3% BSA/0.1% FBS/PBS++. Cells were washed 2 x with ice cold 1X PBS and harvested for RNA by RNeasy Mini Kit (Qiagen). Expression of HA was quantified by qRT-PCR with published primers[28]. Data shown is relative to the HA expression of cells transduced with fluc and bound with 50 MOI IAV.

**IAV minigenome assay.** The influenza A virus minigenome plasmids (pCAGGS-WSN-PB1, pCAGGS-WSN-PB2, pCAGGS-WSN-PA, pCAGGS-WSN-NP, pCAGGS-empty, and pPolI-Luc-GFP) have been described previously and were kindly provided by H. Hoffmann and P. Palese[27]. LY6E KO and WT U2OS were plated at 2.5 x $10^4$ cells per well in a 48-well plate. The next day cells were transfected with the influenza A (A/WSN/33 strain) minigenome constructs pCAGGS expressing PB1, PB2, PA (25 ng each), NP (50 ng), the influenza virus-specific RNA polymerase I driven firefly-GFP dual reporter (pPolI-Luc-GFP) (37.5 ng), and a RNA polymease II driven Renilla luciferase reporter pRLTK (Promega) (25 ng). In control wells, NP was replaced with pCAGGS-empty vector. Twenty-four hours post-transfection, cells were harvested and firefly luciferase and Renilla luciferase activity were assayed on a Berthold luminometer with the Dual Luciferase Assay kit (Promega).

**IAV internalization assay.** Biotin labeling of IAV and the subsequent internalization assay were performed as previously described[28]. In brief, concentrated IAV stocks were diluted to 1 mg/mL viral protein and labeled with sulfo-NHS-SS-biotin (Fisher). Labeled virus was purified by ultracentrifugation through a 30% sucrose cushion. Efficiency of labeling was determined using Pierce Biotin Quantitation Kit. LY6E KO and U2OS cells were incubated in suspension with biotinylated IAV (MOI 10) at 4 °C to promote attachment. Cells were shifted to 37 °C to permit endocytosis. At the indicated timepoints, 15 mM TCEP was added for 15 min at 4 °C to cleave the biotin tag, and then cells were fixed with 1% PFA. Cells were permeabilized with 0.5% saponin, stained for 30 min with 1 µg/mL streptavidin conjugated to AlexaFluor488, and fluorescence intensity was quantified by flow cytometry.

**IAV endosomal escape assay.** R18 labeling of IAV and the subsequent endosomal escape assay were performed as previously described[28]. In brief, IAV pelleted

through a 30% sucrose cushion was diluted to 100 µg/mL was labeled with 7.2 µM rhodamine B (R18). The labeled virus was filtered through a 0.22 µM filter and purified by ultracentrifugation on a 30–50% sucrose gradient. LY6E and KO U2OS were incubated at 4 °C with R18-labeled IAV (MOI 10) to permit attachment. Unbound virus was removed, and cells were shifted to 37 °C. At the indicated time points, 4% PFA was added to fix cells. Fluorescence intensity of the dequenched R18 signal was quantified by flow cytometry.

**IAV acid bypass assay.** The acid bypass assay was performed as previously described by others[63]. In brief, U2OS or STAT1[−/−] fibroblasts were chilled on ice in culture media then incubated for 1 h at 4 °C with concentrated IAV (5 MOI) to allow binding. The cells were washed and incubated for 10 min in either 1X PBS at pH 5.5 or 1X PBS at pH 7.2. PBS was aspirated, replaced with warmed 10% FBS/0.1 mM NEAA/RPMI or DMEM, and cells were shifted to 37 °C. To block spread, bafilomycin A1 at a final concentration of 5 nM was added to the cells 2 h after the temperature shift. Cells were harvested 24 h after the temperature shift, permeabilized, and stained for NP. Percent infection was quantified by flow cytometry.

**IAV uncoating assay.** STAT1[−/−] fibroblasts expressing LY6E or an empty control vector were incubated with concentrated IAV (25 MOI) on ice for 1 h at 4 °C to allow binding. The virus was aspirated and warm media (0.3% BSA/0.1% FBS/1 mM cycloheximide/0.1 mM NEAA/RPMI) was added to cells prior to shifting to 37 °C. Bafilomycin A1 at a final concentration of 5 nM was also added as a control when indicated. Cells were harvested at the indicated time points, fixed, permeabilized, and stained for M1 (1:25 HB64, a kind gift from Andrew Pekosz, Johns Hopkins University) with an AlexaFluor488 secondary. The percentage of M1-positive cells was quantified by flow cytometry.

**IAV NP/nucleus co-localization by ImageStream.** Stable STAT1[−/−] fibroblasts were plated at 7.2 x $10^5$ cells per well on six-well plates that had been previously coated overnight with poly-lysine (10 mg/mL). Plates were equilibrated to 4 °C on ice for 5 min prior to addition of 25 MOI concentrated IAV in 0.3% BSA/0.1% FBS/PBS++. Cells were incubated on ice with virus for 40 min. After the incubation, virus was aspirated and complete media (10% FBS/0.1 mM NEAA/RPMI) was added back and cells were shifted to 37 °C. Bafilomycin A1 was added to a final concentration of 5 nM at indicated time points. After 2 h at 37 °C, cells were trypsinized and stained for NP-AlexaFluor488 as described above. The day of analysis, nuclei were stained with DRAQ5 (Life Technologies) at 20 µM final concentration. A minimum of 1 x $10^4$ in focus cells were collected per sample using Amnis ImageStreamX (Millipore). Analysis of this data was carried out using IDEAS Software (Millpore). To calculate nuclear localization of NP, the mask function was first used to define nuclei based on DRAQ5 staining. We applied the similarity feature, which is the log transformed Pearson's Correlation Coefficient, to measure the degree to which DRAQ5 staining and NP-AF488 signal are linearly correlated pixel by pixel within the masked nuclear region. The resulting similarity value is the NP/nuclear localization score.

**Transferrin uptake assay.** LY6E KO and WT U2OS were plated on a six-well plate at 2 x $10^5$ per well. Cells were incubated for 6 h at 37 °C to allow adherence to the plate. Cells were washed twice with PBS++++ (1 mM CaCl₂/1 mM MgCl₂/0.2% BSA/5 mM glucose/1X PBS pH 7.4) and incubated on ice for 10 min. pHrodo™ Red transferrin Conjugate (Fisher, P35376) was diluted in cold PBS++++ and incubated with cells for 10 min on ice. Cells were shifted to 37 °C for 10 min to allow uptake. Cells were then harvested on ice and the percentage of transferrin-positive cells was determined by flow cytometry.

**Cholera toxin uptake assay.** STAT1[−/−] fibroblasts stably expressing LY6E, empty vector, or CAV1 were plated at 1.5 x $10^5$ cells per well on a poly-lysine coated 24-well plate. The next day, cells were washed twice with cold PBS. Cholera Toxin Subunit B (Recombinant), Alexa Fluor 488 Conjugate (Fisher, C34775) was diluted to 1 µg/mL in PBS++++ and incubated with cells on ice for 30 min. Cells were shifted to 37 °C to allow internalization for 30 min. After incubation, cells were washed three times with cold acid (0.2 M acetic acid/0.2 M NaCl, pH 2.0) for 1 min per wash. Cells were then harvested and the percentage of cholera toxin-positive cells was determined by flow cytometry.

**Dextran uptake assay.** STAT1[−/−] fibroblasts stably expressing LY6E or fluc were incubated with 0.5 mg/mL 70,000 anionic dextran, Oregon Green 488 (Fisher, D7173) diluted in complete RPMI media for 15 min at 37 °C. Cells were harvested on ice and percentage of dextran-positive cells was determined by flow cytometry.

**Molecular phylogenetic analysis.** LY6/uPAR ortholog sequences were selected based on similarity to LY6E, presence of a predicted GPI anchor, and possession of a single three-finger protein motif. The evolutionary history was inferred by using the maximum likelihood method based on the JTT matrix-based model[64]. The tree with the highest log likelihood (−3984.60) is shown. The percentage of trees in which the associated taxa clustered together is shown next to the branches. Initial trees for the heuristic search were obtained automatically by applying Neighbor-

Join and BioNJ algorithms to a matrix of pairwise distances estimated using a JTT model, and then selecting the topology with superior log likelihood value. The tree is drawn to scale, with branch lengths measured in the number of substitutions per site. The analysis involved 11 amino acid sequences. There were a total of 212 positions in the final dataset. Evolutionary analyses were conducted in MEGA7[65].

**Determining amino acid sequence identity and similarity.** Protein sequences of select LY6/uPAR family members were obtained from UniProt[66]. LY6E ortholog nucleotide sequences were obtained from GenBank (NCBI) and translated to obtain the protein sequence using MEGA7[65]. Both analyses were performed using the Ident and Sim feature of the Sequence Manipulation Suite[67].

**Alignment of LY6E orthologs to determine conserved residues for alanine mutagenesis.** Protein sequences of select LY6E orthologs were obtained from UniProt[66]. Clustal Omega was used to align the sequences[68].

**Statistical analyses.** Statistical analyses were performed using GraphPad Prism version 7.02. Individual statistical tests are specified within the figure legends. All statistical analyses were performed prior to normalization. For data with two groups, two-tailed $t$-tests were used under the assumption of normality. Grouped data with more than two groups were analyzed by analysis of variance (ANOVA) under assumption of normality.

## Data availability

The authors declare that the data supporting the findings of this study are available within the article and its Supplementary Information files or are available on request. mRNA-seq data has been deposited in GEO and is available with the accession code GSE111958.

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

## Acknowledgements

We thank the UT Southwestern Flow Cytometry and Next Generation Sequencing Core Facilities for technical support. We thank M. Daugherty, N. Alto, S. Cherry, and the Schoggins lab for helpful discussions. This work was supported in part by NIH grants DK095031 and AI117922 (J.W.S.). Support was also obtained from the UT Southwestern Endowed Scholars Program and the Clayton Foundation (J.W.S.). Additional support was obtained from T32 Training Grants AI005284 (K.B.M.) and AI007520 (N.R.R.), and National Science Foundation Graduate Research Fellowship Program grant 2016217834 (I.N.B.). The funders had no role in study design, data collection and interpretation, or the decision to submit the work for publication.

## Author contributions

K.B.M. and J.W.S. designed the study. All authors performed experiments and contributed to data analysis. K.B.M. and I.N.B. performed evolutionary analyses. K.B.M., I.N.B., and J.W.S. wrote the manuscript.

## Additional information

**Competing interests:** The authors declare no competing interests.

