## [Peer Review File · Nature Communications]

Reviewers' comments:

Reviewer #1 (Remarks to the Author):

The current manuscript is an extension of previous work from this group identifying interferon-stimulated genes (ISGs) as both negative and positive regulators of virus infection. Specifically, lymphocyte antigen 6 complex, locus E (LY6E) was previously identified to augment infection of multiple enveloped viruses. The current manuscript extends those findings to show that LY6E ectopic expression enhances infection particularly of flaviviruses and influenza A virus (IAV) in fibroblasts and U2OS cells but not A549 or BHK cells suggesting cell-type specificity. Knock-out of LY6E expression using CRISPR/Cas gene editing or siRNA reduced infection of IAV in THP-1 monocytic cells and U2OS cells. LY6E knock-out did not change the responsiveness of cells to type I IFN, or change gene transcription profiles by RNAseq suggesting the effects of LY6E on virus infection were direct. In the context of yellow fever virus (YFV), overexpression of LY6E did not significantly affect virus binding to cells, viral RNA replication or gene translation, or packaging and release of infectious virus. This suggests that LY6E acts at a step after virus binding but before translation of the virus genome. The authors use a bafilomycin A inhibition of endosomal acidification assay to measure virus entry of YFV, and examine IAV nucleoprotein localization and movement from the cell periphery to the nucleus to infer that the major replication step affected by LY6E is viral entry. Finally, they show that Ly6E function in enhancement of virus replication is conserved across diverse species, including non-human primate, flying fox bat species and rodents. Thus the manuscript confirms the role of LY6E as an ISG that enhances replication of important human viral pathogens, and suggests that it acts to augment early steps in the virus replication cycle associated with virus entry.

Specific comments:

1. The manuscript is largely descriptive and therefore reveals minimal mechanistic insight as to how LY6E functions to augment virus infection besides potentially enhancing virus entry. Increased mechanistic insight is important since a function for LY6E as a positive regulator of virus infectivity has been previously reported in multiple publications by this group. If the effect of LY6E is on virus endocytosis, does LY6E affect endocytosis generally? Can this be shown using uptake of fluorescent beads, particularly in THP-1 cells, for example? Alternatively, transferrin uptake or uptake of fluorescent dextran could be examined and potentially provide considerable insight as to the role of LY6E in infection. Without further understanding of how LY6E functions, the work represents a minimal advance in understanding of this phenomenon.
2. The data using BafA as an entry inhibitor are not convincing. In addition to endosomal acidification, BafA will suppress lysosomal acidification, including pathways of autophagy that viruses often use or inhibit. In addition, the BafA inhibitor appears to left on the cells for 48h of infection and a measurement of 48h is examining the entire replication cycle in the presence of BafA and not just virus entry. BafA treatment of cells for that amount of time is generally detrimental to cellular function and it is possible that a role for LY6E in protection from cell death might contribute to the differences observed. Therefore, an assay designed to specifically examine virus entry would be more informative.

3. The general conclusion that ISGs that enhance infectivity is novel or represents a 'new class' of ISGs is an overstatement. This phenomenon has been described multiple times previously in addition to the Axl examine discussed by the authors, including SOCS3 in the context of Ebola virus, and viperin enhancement of HCMV. This overselling of the manuscript should be amended.

4. Figure 3F depicts a very low percentage of cells infected at 2-4%. This low level of infection (implying a low level of virus egress) may make it difficult to observe meaningful differences between conditions. Can this experiment be optimized further to increase the linear range of the readout and therefore increase confidence of the result?

5. LY6E overexpression is the only experimental condition with no other controls shown for protein overexpression (except for IRF1 in one experiment). For example, inclusion of a GPI-anchorless LY6E might be an ideal control for these studies and simultaneously provide information on role of cell surface expression of LY6E in enhancement of virus infectivity.

Other comments:

1. It is not clear how the NP/nucleus similarity score is generated. Is it derived from a vector drawn across each cell followed by generation of the ratio of cytosolic to nuclear localization? Please clarify.

2. Lines 368-373: The conclusion here reads like there could be two effects of LY6E, virus entry and vRNP trafficking. However, the effect on vRNP trafficking could equally be due to effects on virus entry and have nothing to do with the actual trafficking to the nucleus. Please clarify what is meant here.

Reviewer #2 (Remarks to the Author):

LY6E was previously identified as an ISG, the expression of which stimulates infection of different enveloped RNA viruses. In the current study, the authors study in more detail the role of the LY6E protein in the infection cycle of various enveloped RNA viruses. The authors confirm their previous results by showing that ectopically expressed LY6E enhances different RNA viruses in different, but not all, cell types tested. In addition, they show that KO of LY6E resulted in reduced cell susceptibility for IAV (but not for other viruses) . LY6E was shown not to affect IFN activity and signaling or cellular transcription. LY6E was also shown not to directly affect attachment, RNA replication or production of YVF (but not for other viruses). LY6E did enhance viral entry of YFV and IAV.

While the experiments were well described and performed, the conclusions are often too generalized. In addition, the authors do not shed much new light on the proviral role of LY6E compared to their previous study. While the data indicate that viral entry is enhanced by this protein, which was to be expected in view of LY6E localization at the cell surface (and probably the endocytic route), it is not clear at which step (signaling, trafficking through the endocytic route, fusion, uncoating?).

Other points:

1. Fig. 1 What could explain the cell-type dependency that is observed in panel D? Did the authors analyze endogenous LY6E levels in these cells?

2. Fig. 1. Why did the authors use STAT1^{-/-} cells?
3. Fig. 2 only shows results for IAV. Conclusions drawn from these experiments are thus only valid for IAV. The authors should change the manuscript accordingly or perform the required experiments.
4. Fig. 3 shows only results for YFV. The authors cannot exclude that LY6E does affect virus attachment and replication for other viruses. This should be explicitly stated or the experiments should be performed.
5. statistics appear ok.

Reviewers' comments:

Reviewer #1 (Remarks to the Author):

The current manuscript is an extension of previous work from this group identifying interferon stimulated genes (ISGs) as both negative and positive regulators of virus infection. Specifically, lymphocyte antigen 6 complex, locus E (LY6E) was previously identified to augment infection of multiple enveloped viruses. The current manuscript extends those findings to show that LY6E ectopic expression enhances infection particularly of flaviviruses and influenza A virus (IAV) in fibroblasts and U2OS cells but not A549 or BHK cells suggesting cell-type specificity. Knock-out of LY6E expression using CRISPR/Cas gene editing or siRNA reduced infection of IAV in THP-1 monocytic cells and U2OS cells. LY6E knock-out did not change the responsiveness of cells to type I IFN, or change gene transcription profiles by RNAseq suggesting the effects of LY6E on virus infection were direct. In the context of yellow fever virus (YFV), overexpression of LY6E did not significantly affect virus binding to cells, viral RNA replication or gene translation, or packaging and release of infectious virus. This suggests that LY6E acts at a step after virus binding but before translation of the virus genome. The authors use a bafilomycin A inhibition of endosomal acidification assay to measure virus entry of YFV, and examine IAV nucleoprotein localization and movement from the cell periphery to the nucleus to infer that the major replication step affected by LY6E is viral entry. Finally, they show that LY6E function in enhancement of virus replication is conserved across diverse species, including non-human primate, flying fox bat species and rodents. Thus the manuscript confirms the role of LY6E as an ISG that enhances replication of important human viral pathogens, and suggests that it acts to augment early steps in the virus replication cycle associated with virus entry.

Specific comments:

1. The manuscript is largely descriptive and therefore reveals minimal mechanistic insight as to how LY6E functions to augment virus infection besides potentially enhancing virus entry. Increased mechanistic insight is important since a function for LY6E as a positive regulator of virus infectivity has been previously reported in multiple publications by this group. If the effect of LY6E is on virus endocytosis, does LY6E affect endocytosis generally? Can this be shown using uptake of fluorescent beads, particularly in THP-1 cells, for example? Alternatively, transferrin uptake or uptake of fluorescent dextran could be examined and potentially provide considerable insight as to the role of LY6E in infection. Without further understanding of how LY6E functions, the work represents a minimal advance in understanding of this phenomenon.

We appreciate this critique and have attempted to incorporate new data to glean mechanistic insight into LY6E function. On the reviewer's suggestion, we examined the effect of LY6E on transferrin uptake in WT and LY6E KO U2OS cells (generated by CRISPR). We observed no difference between the two cell types (FIGURE 5G). These data suggest that the general process of clathrin-mediated uptake is not enhanced by LY6E. We also tested other ligands that enter cells through independent processes, including dextran and cholera toxin (FIGURE 5H-I) using cells overexpressing LY6E. These were also unaffected by LY6E. We assessed virus internalization with biotinylated influenza A virus (IAV) and found that LY6E had no effect (FIGURE 5J). Further mechanistic study to dissect the affected step of viral entry revealed that LY6E affects a step after endosomal escape (FIGURE 5K). Using a simple acid bypass assay, we demonstrate that forcing viral fusion to occur at the plasma membrane did not lead to loss of viral enhancement by LY6E (FIGURE 5L-M). Since all steps up until uncoating are bypassed in this assay, we conclude that LY6E affects a step after endosomal escape which is likely uncoating.

In addition to these new studies, we have added an extensive alanine scanning mutagenesis of LY6E, the results of which implicate the flexible finger domains of the protein as critical for the viral enhancement phenotype (FIGURE 7). We also attempted to cleave LY6E from the cell surface using GPI-specific phospholipase C, but were unable to fully cleave over-expressed protein relative to GPI-anchored GFP. We infected PLC-treated LY6E and control cells and observed no changes in infection (included in rebuttal only, see below).

Rebuttal FIGURE 1. Phospholipase C treatment. (A) *STAT1*^{-/-} fibroblasts expressing GFP-HA bearing the signal peptide and GPI-anchor of LY6E were treated for 1 hour with 1% (0.5 U/mL) phospholipase C (PLC) or vehicle. MFI of GFP signal was assessed by flow cytometry. (B) Immunofluorescent imaging of (A). (C) *STAT1*^{-/-} fibroblasts expressing LY6E were treated as in (A) with PLC or vehicle. Cells were harvested, stained for LY6E or with an isotype control, and MFI of the fluorescent secondary was quantified by flow cytometry. (D) *STAT1*^{-/-} fibroblasts expressing LY6E or fluc control were treated as described in (A) with PLC. Cells were then infected with YFV-17D-Venus (0.4 MOI, 24h). Cells were harvested and percent infection was quantified by flow cytometry.

2. The data using BafA as an entry inhibitor are not convincing. In addition to endosomal acidification, BafA will suppress lysosomal acidification, including pathways of autophagy that viruses often use or inhibit. In addition, the BafA inhibitor appears to left on the cells for 48h of infection and a measurement of 48h is examining the entire replication cycle in the presence of BafA and not just virus entry. BafA treatment of cells for that amount of time is generally detrimental to cellular function and it is possible that a role for LY6E in protection from cell death might contribute to the differences observed. Therefore, an assay designed to specifically examine

virus entry would be more informative.

The reviewer makes an excellent point regarding the problems with long term BafA treatment. In the entry studies with YFV (FIGURE 5B-C), we indeed left the BafA on the cells until viral replication was assessed at 48 h. However, for the influenza entry experiments using ImageStream, we briefly pre-treated the cells with BafA and harvested cells for ImageStream analysis within 2 hours of infection to strictly zero in on entry. We have modified the text and Methods to make this critical experimental detail more clear. In addition to these studies, we also showed that direct electroporation of viral RNA, which bypasses entry, leads to the same levels of viral replication whether LY6E is expressed or not. Since the electroporated RNA isolates viral translation and replication, which are the same in both WT and LY6E cells, this experiment further supports the conclusion that the effect of LY6E is on entry. This data set was originally placed in FIGURE 4, but we have moved it to FIGURE 5 to better support the entry mechanism.

3. The general conclusion that ISGs that enhance infectivity is novel or represents a 'new class' of ISGs is an overstatement. This phenomenon has been described multiple times previously in addition to the Axl examine discussed by the authors, including SOCS3 in the context of Ebola virus, and viperin enhancement of HCMV. This overselling of the manuscript should be amended.

The abstract and discussion has been revised in consideration of this comment.

4. FIGURE 3F depicts a very low percentage of cells infected at 2-4%. This low level of infection (implying a low level of virus egress) may make it difficult to observe meaningful differences between conditions. Can this experiment be optimized further to increase the linear range of the readout and therefore increase confidence of the result?

The experiment mentioned by the reviewer used a YFV-17D-Venus RNA, which did not yield high titers in under the experimental conditions used. We therefore repeated the experiment with non-reporter YFV-17D RNA and collected supernatants for plaque assay, which had an average titer of approximately 1e6 PFU/mL, which is more typical of YFV virus production (FIGURE 4E).

5. LY6E overexpression is the only experimental condition with no other controls shown for protein overexpression (except for IRF1 in one experiment). For example, inclusion of a GPI-anchorless LY6E might be an ideal control for these studies and simultaneously provide information on role of cell surface expression of LY6E in enhancement of virus infectivity.

Our data may have not been clear, but we always ectopically express firefly luciferase or empty cassette from a lentiviral vector as a control for lentivirus-expressed LY6E. The fluc control was well-validated in Schoggins, Nature 2011. We also have previously published use of an empty vector cassette as a control in Rinkenberger, mBio 2018. Granted, while luciferase helps control for effects of lentiviral transduction and general protein overexpression, it is not a control to glean insight into LY6E function. In lieu of a GPI-anchorless LY6E, our new alanine-scanning mutagenesis (FIGURE 7A-D) includes block or point mutants in LY6E that readily express, but do not confer viral enhancement. These constructs should represent strong controls for overexpression, while also providing insight into LY6E function.

Other comments:

1. It is not clear how the NP/nucleus similarity score is generated. Is it derived from a vector drawn

across each cell followed by generation of the ratio of cytosolic to nuclear localization? Please clarify.

We have added additional information on the ImageStream analysis in the Methods. The similarity score/Pearson's correlation coefficient were also added back to the histograms in FIGURE 5E for further clarification.

In brief: "Analysis of this data was carried out using IDEAS Software (Millipore). To calculate nuclear localization of NP, the mask function was first used to define nuclei based on DRAQ5 staining. We applied the similarity feature, which is the log transformed Pearson's Correlation Coefficient, to measure the degree to which DRAQ5 staining and NP-AF488 signal are linearly correlated pixel by pixel within the masked nuclear region. The resulting similarity value is the NP/nuclear localization score."

2. Lines 368-373: The conclusion here reads like there could be two effects of LY6E, virus entry and vRNP trafficking. However, the effect on vRNP trafficking could equally be due to effects on virus entry and have nothing to do with the actual trafficking to the nucleus. Please clarify what is meant here.

In the discussion, we mentioned the ImageStream analysis to argue that LY6E appears to affect the rate of entry in addition to increasing the percentage of cells that are infected. The wording for this part of the discussion has been revised in consideration of these comments. In addition, we clarified in the Results section that LY6E is unlikely to affect trafficking to the nucleus, as YFV entry is also enhanced and YFV replicates in the cytoplasm.

Reviewer #2 (Remarks to the Author):

LY6E was previously identified as an ISG, the expression of which stimulates infection of different enveloped RNA viruses. In the current study, the authors study in more detail the role of the LY6E protein in the infection cycle of various enveloped RNA viruses. The authors confirm their previous results by showing that ectopically expressed LY6E enhances different RNA viruses in different, but not all, cell types tested. In addition, they show that KO of LY6E resulted in reduced cell susceptibility for IAV (but not for other viruses). LY6E was shown not to affect IFN activity and signaling or cellular transcription. LY6E was also shown not to directly affect attachment, RNA replication or production of YFV (but not for other viruses). LY6E did enhance viral entry of YFV and IAV.

While the experiments were well described and performed, the conclusions are often too generalized. In addition, the authors do not shed much new light on the proviral role of LY6E compared to their previous study. While the data indicate that viral entry is enhanced by this protein, which was to be expected in view of LY6E localization at the cell surface (and probably the endocytic route), it is not clear at which step (signaling, trafficking through the endocytic route, fusion, uncoating?).

We thank the reviewer for the positive comments on the experimental approach. With respect to generalized conclusions, a similar point made by Reviewer 1, we have modified the manuscript throughout to remove generalizations and be more specific. We have also added new mechanistic data, including a viral endosomal escape assay, acid bypass assay, transferrin uptake assay, and a structure-function analysis via alanine scanning mutagenesis.

Other points:

1. Fig. 1 What could explain the cell-type dependency that is observed in panel D? Did the authors analyze endogenous LY6E levels in these cells?

We examined endogenous and IFN-induced levels of LY6E by Western blot and were unable to conclude a link to cell-type dependency of the LY6E phenotype (FIGURE S1A-C).

2. Fig. 1. Why did the authors use STAT1^{-/-} cells?

Most experiments were done in STAT1^{-/-} cells because the original screen identifying the viral enhancement phenotype of LY6E was done in this cell type. We were originally concerned that some ISGs would simply impact IFN signaling, and we wanted to limit our screening hits to those ISGs that specifically affected viral infection in the absence of STAT1-dependent IFN signaling. We demonstrate that the phenotype is STAT1 independent in FIGURE S3 and by showing enhancement in STAT1-sufficient human fibroblasts (FIGURE S1A). Further justification has been added to FIGURE 1 text.

3. Fig. 2 only shows results for IAV. Conclusions drawn from these experiments are thus only valid for IAV. The authors should change the manuscript accordingly or perform the required experiments.

Originally FIGURE 2 (now FIGURE 3) had the results of infecting KO THP-1 and KO U2OS with IAV, but FIGURE S1 contained results for YFV and ONNV as well. To address this concern, we compiled the data in the KO cells for all viruses into FIGURE 3.

4. Fig. 3 shows only results for YFV. The authors cannot exclude that LY6E does affect virus attachment and replication for other viruses. This should be explicitly stated or the experiments should be performed.

FIGURE 3, which is now FIGURE 4, indeed only uses YFV as a model virus, as we possess tools to look at attachment and replication of this virus. We have amended the section for FIGURE 4 to reflect that we only used YFV and added a statement confirming that these studies need to be extended to other viruses.

5. statistics appear ok.

Reviewers' comments:

Reviewer #1 (Remarks to the Author):

The authors have added additional data to support the conclusion that LY6E augments viral entry at a step following endosomal escape but before RNA replication, potentially at virus uncoating. The data are certainly improved, and more precisely indicate the point in virus infection impacted by LY6E.

Figure 7D: L36A does not appear to be specifically required for viral enhancement. It seems to be required for LY6E stability or expression. These conclusions need to more accurately reflect the data.

Minor:

Figure 2C should be 3C (line 165)

Reviewer #2 (Remarks to the Author):

My main criticism was and still is that the conclusions are often too generalized. The authors claim to have addressed this issue throughout the manuscript. The abstracts still reads, however, "mechanistically, we narrowed the enhancing effect of LY6E to a post-attachment viral entry step after endosomal escape but prior to onset of replication". In view also of the statement in the discussion section "LY6E may also enhance different parts of viral entry for different viruses, as a recent study using HIV-1 identified an effect on viral internalization end Env-mediated membrane fusion", I am not sure about the correctness of the statement in the abstract. I am also not convinced that the authors have addressed this point throughout the manuscript.

This is for example illustrated by the sentence on page 11: "Since the bafA assay relied on YFV gene expression at a late time point (48 hours), we next wanted to monitor viral entry more directly by tracking incoming virus. using ImageStream flow cytometry, we visualized the effect of LY6E on IAV nucleoprotein (NP) trafficking at multiple time points up to 2 hours after synchronized infection." The authors thus use the observation they have made for one virus as a rationale to perform a follow-up experiment with another virus.

Furthermore, the authors show for YFV that replication and translation is not affected, but do not show that this is also the case for IAV. Nevertheless, the interpretation of their results they assume that RNA replication/translation for IAV is not affected. As LY6E absence/presence did not affect endocytic uptake or membrane fusion, but did affect virus replication after low-pH induced fusion, a role for LY6E in IAV replication cannot yet be excluded.

The authors are thus advised to completely analyse the different steps of the virus-life cycle for YFV and IAV and then still to be cautious about over interpretation of these findings for other viruses.

Other remarks:

The authors suggest that uncoating must be affected, but they do not show convincing proof. These assays are available for IAV (based on antibody staining of M1).

The alanine mutagenesis is little informative about the mechanism of LY6E-mediated viral enhancement. Furthermore, is the same residue important for the LY6E-effect in other virus infections?

Several viruses are not affected by absence/presence of LY6E. Can the authors speculate to what extent these viruses differ from the LY6E-dependent viruses?

Standard deviations are lacking in Fig. 5F.

Reviewers' comments:

Reviewer #1 (Remarks to the Author):

The authors have added additional data to support the conclusion that LY6E augments viral entry at a step following endosomal escape but before RNA replication, potentially at virus uncoating. The data are certainly improved, and more precisely indicate the point in virus infection impacted by LY6E.

We appreciate the reviewer's comments on our improved manuscript.

Figure 7D: L36A does not appear to be specifically required for viral enhancement. It seems to be required for LY6E stability or expression. These conclusions need to more accurately reflect the data.

We modified how we presented our conclusions for this data, adding that L36 appears important for both LY6E expression and viral enhancement (lines 375-376). Increasing expression of L36A to match that of WT LY6E did not rescue the viral phenotype (Supplementary Fig. 5d), so we stand by our conclusion that the lack of phenotype is not sufficiently explained by a decrease in expression or stability.

Minor:

Figure 2C should be 3C (line 165)

We corrected this typo.

Reviewer #2 (Remarks to the Author):

My main criticism was and still is that the conclusions are often too generalized. The authors claim to have addressed this issue throughout the manuscript. The abstract still reads, however, "mechanistically, we narrowed the enhancing effect of LY6E to a post-attachment viral entry step after endosomal escape but prior to onset of replication". In view also of the statement in the discussion section "LY6E may also enhance different parts of viral entry for different viruses, as a recent study using HIV-1 identified an effect on viral internalization and Env-mediated membrane fusion", I am not sure about the correctness of the statement in the abstract. I am also not convinced that the authors have addressed this point throughout the manuscript.

This is for example illustrated by the sentence on page 11: "Since the bafA assay relied on YFV gene expression at a late time point (48 hours), we next wanted to monitor viral entry more directly by tracking incoming virus. Using ImageStream flow cytometry, we visualized the effect of LY6E on IAV nucleoprotein (NP) trafficking at multiple time points up to 2 hours after synchronized infection." The authors thus use the observation they have made for one virus as a rationale to perform a follow-up experiment with another virus.

Furthermore, the authors show for YFV that replication and translation is not affected, but do not show that this is also the case for IAV. Nevertheless, the interpretation of their results they assume that RNA replication/translation for IAV is not affected. As LY6E absence/presence did not affect endocytic uptake or membrane fusion, but did affect virus replication after low-pH induced fusion, a role for LY6E in IAV replication cannot yet be excluded.

The authors are thus advised to completely analyse the different steps of the virus-life cycle for YFV and IAV and then still to be cautious about over interpretation of these findings for other viruses.

We appreciate this critique and have rewritten the parts of the manuscript mentioned above as well as any other statements that we also observed to be "too generalized." Our experiments for YFV and IAV are now presented as two separate figures. We also added viral attachment and replication/translation (minigenome) experiments for IAV, as suggested by the reviewer (new Fig. 5a, 5b).

Other remarks:

The authors suggest that uncoating must be affected, but they do not show convincing proof. These assays are available for IAV (based on antibody staining of M1).

We thank the reviewer for this important experimental suggestion. We obtained the HB64 mouse monoclonal antibody that detects M1 antigen from uncoated IAV at an improved level over virus that has not undergone uncoating. Indeed, uncoated IAV led to an increased M1 signal as measured by flow cytometry, whereas treatment with bafilomycin A1 to block endosomal escape reduced the percentage of M1-positive events. All uncoating experiments were done with the addition of cycloheximide to prevent synthesis of nascent M1. Ectopic LY6E expression enhanced uncoating as measured by staining for M1 (new Fig. 5g). These data are now better able to support the conclusion by using a more direct assay.

The alanine mutagenesis is little informative about the mechanism of LY6E-mediated viral enhancement. Furthermore, is the same residue important for the LY6E-effect in other virus infections?

We would contend that the alanine scanning mutagenesis is more geared toward studying structure-function relationships rather than defining molecular mechanisms. We hope this data serves as a jumping off point for future studies that probe the structure in more depth. As suggested, we tested our L36A mutant against IAV and observed that it also results in a loss of the LY6E enhancement phenotype (new Fig. 7e).

Several viruses are not affected by absence/presence of LY6E. Can the authors speculate to what extent these viruses differ from the LY6E-dependent viruses?

This is an important point, and we added some modest speculation about this observation in our discussion. In brief, viruses have different dependencies on host cell machinery. But, without knowing the precise part of viral entry that is affected for other viruses, adding detailed speculation to this manuscript may be overstepping our conclusions. Unfortunately, an uncoating assay homologous to the M1 stain used for IAV is not available for YFV, nor has any work been done by other groups to thoroughly define the uncoating machinery required by YFV.

Standard deviations are lacking in Fig. 5F.

Originally, we showed mean scores from a single replicate for the ImageStream assay. While we observed increased nuclear translocation relative to control cells in all our replicates, the magnitude of change varied significantly from experiment to experiment. We thus have chosen to present all three replicates separately and without statistical analyses.

REVIEWERS' COMMENTS:

Reviewer #2 (Remarks to the Author):

The authors performed additional experiments and adapted the manuscript as requested.

REVIEWERS' COMMENTS:

Reviewer #2 (Remarks to the Author):

The authors performed additional experiments and adapted the manuscript as requested.

We thank reviewer #2 for their critiques throughout the review process. Their rigorous comments have contributed to an immensely improved manuscript.